# THE ADAPTIVE COMPLEXITY OF PARALLELIZED LOG-CONCAVE SAMPLING

**Huanjian Zhou**[1,2] **Baoxiang Wang** [3,4]**, Masashi Sugiyama**[2,1]*

[1]The University of Tokyo [2]RIKEN AIP

[3]The Chinese University of Hong Kong Shenzhen [4]Vector Institute

## ABSTRACT

In large-data applications, such as the inference process of diffusion models, it is desirable to design sampling algorithms with a high degree of parallelization. In this work, we study the adaptive complexity of sampling, which is the minimum number of sequential rounds required to achieve sampling given polynomially many queries executed in parallel at each round. For unconstrained sampling, we examine distributions that are log-smooth or log-Lipschitz and log strongly or non-strongly concave. We show that an almost linear iteration algorithm cannot return a sample with a specific exponentially small error under total variation distance. For box-constrained sampling, we show that an almost linear iteration algorithm cannot return a sample with sup-polynomially small error under total variation distance for log-concave distributions. Our proof relies upon novel analysis with the characterization of the output for the hardness potentials based on the chain-like structure with random partition and classical smoothing techniques.

## 1 INTRODUCTION

We study the problem of sampling from a target distribution on $\mathbb{R}^d$ given query access to its unnormalized density, which is fundamental in many fields such as Bayesian inference, randomized algorithms, and machine learning (Marin et al., 2007; Nakajima et al., 2019; Robert et al., 1999). Recently, significant progress has been made in developing sequential algorithms for this problem inspired by the extensive optimization toolkit, particularly when the target distribution is log-concave (Chewi et al., 2020; Jordan et al., 1998; Lee et al., 2021; Wibisono, 2018; Ma et al., 2019).

The algorithms underlying the above results are highly sequential and fail to fully exploit contemporary parallel computing resources such as multi-core central processing units (CPUs) and many-core graphics processing units (GPUs). While model-specific algorithmic subroutines such as log-likelihood and log-likelihood gradient evaluations sometimes admit parallelization (Holbrook et al., 2021a;b), the algorithms' generally sequential nature can lead to under-utilization of increasingly widespread hardware (Brockwell, 2006).

A convenient metric for parallelism in black-box oracle models is adaptivity, which was recently introduced in submodular optimization to quantify the information-theoretic complexity of black-box optimization in a parallel computation model (Balkanski and Singer, 2018a). Informally, the adaptivity of an algorithm is the number of sequential rounds it makes when each round can execute polynomially many independent queries in parallel. In the past several years, there have been breakthroughs in the study of adaptivity in optimization (Balkanski and Singer, 2018a;b; 2020; Bubeck et al., 2019; Diakonikolas and Guzmán, 2019; Chakrabarty et al., 2023; 2024; Garg et al., 2021; Li et al., 2020).

Although the adaptive complexity of optimization is well understood, we have a very limited understanding of the adaptive complexity of sampling. Existing results are only on query complexity of the low-dimensional sampling Chewi et al. (2022b; 2023a;b). This motivates us to study the adaptive complexity of log-concave samplers. In this paper, we give the first lower bounds for the

---

*Corresponding to: Huanjian Zhou, Baoxiang Wang, Masashi Sugiyama

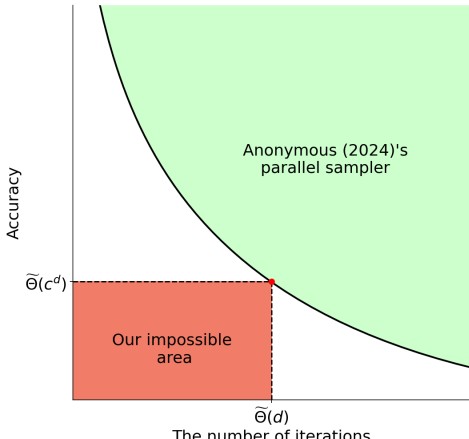

Figure 1: Comparison with existing parallel methods for strongly log-concave and log-smooth distributions

parallel runtime of sampling in high dimensions and high accuracy regimes[1]. We study two types of log-concave samplers: unconstrained samplers and box-constrained samplers.

**Lower bound for unconstrained samplers.** We first present the lower bound for unconstrained samplers with very high accuracy. Specifically, for sufficiently large dimensions, an almost linear iteration adaptive sampler fails to return a sample with a specific exponentially small accuracy for (i) strongly log-concave and log-smooth distributions (ii) log-concave and log-smooth or log-Lipschitz distributions, and (iii) composite distributions (see Theorem 4.1, Theorem 4.4, and Theorem 4.5 respectively). This is the first lower bound to the best of our knowledge for both deterministic adaptive samplers and randomized ones for unconstrained distributions.

Take the strongly log-concave and log-smooth sampler as an example. Let $d$ and $\varepsilon$ denote the dimension and accuracy, respectively. For any parallel samplers running in $\widetilde{\Theta}(d)$ iterations, we prove the accuracy is always $\widetilde{\omega}(c^d)$ with some constant $c < 1$. Conversely, improving accuracy beyond our current bounds necessitates an increase in iterations. These correspond to the impossible region (the red rectangle) in Figure 1. As a comparison, Anari et al.'s algorithm returns a sample with $\varepsilon$ accuracy under total variation distance within $\mathcal{O}(\log^2(d/\varepsilon))$ iterations and makes $\mathcal{O}(d)$ queries in each iteration (Anari et al., 2024). The iteration complexity was later improved to $\mathcal{O}(\log(d/\varepsilon))$ a logarithmic increase in the number of queries per iteration Anonymous (2024), corresponding to the green area in Figure 1. Our lower bound matches this upper bound, emphasizing the critical role of the term $\log(1/\varepsilon)$ and ruling out the possibility of designing super-exponentially converging sampling algorithms, even for sequential methods. For weakly log-concave samplers, the algorithm by Fan et al. requires $\widetilde{\mathcal{O}}(\log^2(1/\varepsilon))$ iterations for log-Lipschitz distributions and $\widetilde{\mathcal{O}}(d^{1/2}\log^2(1/\varepsilon))$ iterations for log-smooth distributions, with $\mathcal{O}(1)$ queries per iteration (Fan et al., 2023). We plug the accuracy of our lower bound in to their guarantees and summarize the comparisons in Table 1.

**Lower bound for box-constrained samplers.** We also extend the result to the box-constrained setting. Sampling from polytope-constrained has a wide range of applications including Bayesian inference and differential privacy Silvapulle and Sen (2011); McSherry and Talwar (2007). The box constraint represents the simplest case, such as in the Bayesian logistic regression problem with the infinity norm. We show that for sufficiently large dimensions, any almost linear-iteration adaptive sampler for box-constrained log-concave distributions fails to return a sample with sup-polynomially small accuracy (see Theorem 5.1).

The best existing algorithm employs the soft Dikin walk to return a lower-accurate even high-accurate sample within $\mathcal{O}(d^3)$ iterations for general polytopes (Mangoubi and Vishnoi, 2023b). However, there also exists a gap compared to our lower bounds. For strongly log-concave and log-smooth

---

[1]Throughout, we use the standard terminology Chewi (2023) low accuracy to refer to complexity results which scale polynomially in $1/\varepsilon$, and the term high accuracy for results which scale polylogarithmically in $1/\varepsilon$, here, $\varepsilon > 0$ is the desired target accuracy.

Table 1: Comparison with existing upper bounds in different levels of accuracy. We note $c$ is some constant such that $c \in (0, 1)$ and $\alpha$ is an arbitrary number satisfying $\alpha = \omega(1)$. We hide the poly-logarithmic term. ♣ For the upper bound of the log-concave sampler, we assume there exists an $\mathcal{O}(d)$-warm start with bounded Chi-square divergence, i.e., $\chi_\pi^2(\rho_0) = \mathcal{O}(\exp(d))$. Here $\mathsf{m}_2$ denotes the second moment. ♠ We only consider the potential functions taking form as $f = f_1 + \|\cdot\|^2/2$ where $f_1$ is convex and Lipschitz.

| Problem type of adaptive sampling | | Upper bounds | Our lower bounds |
|---|---|---|---|
| Unconstrained dist. with $\varepsilon = \Theta(c^d)$ | strongly log-concave + log-smooth | $\widetilde{\mathcal{O}}(d)$ (Anonymous, 2024) | $\widetilde{\Omega}(d)$ |
| | log-concave + log-smooth | $\widetilde{\mathcal{O}}(\mathsf{m}_2 d^{5/2})$ (Fan et al., 2023) ♣ | $\widetilde{\Omega}(d)$ |
| | log-concave + log-Lipschitz | $\widetilde{\mathcal{O}}(\mathsf{m}_2 d^2)$ (Fan et al., 2023) ♣ | $\widetilde{\Omega}(d)$ |
| | composite ♠ | $\widetilde{O}(d^{5/2})$ (Fan et al., 2023) | $\widetilde{\Omega}(d)$ |
| box-constrained dist. with $\varepsilon = \Theta(d^{-\alpha})$ | log-concave + log-smooth | $\widetilde{\mathcal{O}}(d^3)$ (Mangoubi and Vishnoi, 2023a) | $\widetilde{\Omega}(d)$ |
| | log-concave + log-Lipschitz | $\widetilde{\mathcal{O}}(d^3)$ (Mangoubi and Vishnoi, 2023a) | $\widetilde{\Omega}(d)$ |
| | strongly-log-concave + log-smooth | $\widetilde{\mathcal{O}}(d)$ (Lee et al., 2021) | - |
| box-constrained dist. with $\varepsilon = \Theta(c^d)$ | log-concave + log-smooth | $\widetilde{\mathcal{O}}(d^3)$ (Mangoubi and Vishnoi, 2023a) | $\widetilde{\Omega}(d)$ |
| | log-concave + log-Lipschitz | $\widetilde{\mathcal{O}}(d^3)$ (Mangoubi and Vishnoi, 2023a) | $\widetilde{\Omega}(d)$ |
| | strongly-log-concave + log-smooth | $\widetilde{\mathcal{O}}(d^4)$ (Lee et al., 2021) | - |

distributions, we can view the box-constrained distribution as a composite distributions. As a result, the proximal sampler can find a high-accurate sample within $\widetilde{\mathcal{O}}(d^4)$ iterations, and a sample with low accuracy within $\widetilde{\mathcal{O}}(d)$ iterations (Lee et al., 2021). But the lower bounds are still unknown. We also summarize it in Table 1.

**Comparision to query complexity.** The current understanding of the query complexity of sampling is notably limited. For general strongly log-concave and log-smooth distributions, investigations have been confined primarily to 1- and 2-dimensional tasks, and only apply to a certain constant accuracy (Chewi et al., 2022b; 2023b). For high-dimensional tasks, even for the Gaussian distribution, studies have only addressed constant accuracy (Chewi et al., 2023b). Also, a lower bound for box-constrained log-concave samplers is conspicuously absent. Our work provides the first response to these limitations, highlighting the critical role of the term $\log(1/\varepsilon)$ and ruling out the feasibility of designing super-exponentially converging sampling algorithms.

**Exponential accuracy requirements with privacy as an example.** In differential privacy, one requires bounds on the infinity-distance (Mangoubi and Vishnoi, 2022) or Wasserstein-infinity distance (Lin et al., 2023) to guarantee pure differential privacy, and total variation (TV), Kullback–Leibler (KL), or Wasserstein bounds are insufficient (Dwork et al., 2014). To achieve these stringent privacy guarantees, Mangoubi and Vishnoi (2022) or Lin et al. (2023) both designed algorithms that first generate an exponentially accurate sample w.r.t. TV, and then convert samples with TV bounds to infinity-distance bounds. Moreover, in rare-event statistics, high-accuracy simulations are required (Shyalika et al., 2023).

## 2 PRELIMINARIES

**Sampling task** Given the potential function $f : \mathcal{D} \to \mathbb{R}$, the goal of the sampling task is to draw a sample from the density $\pi_f = Z_f^{-1} \exp(-f)$, where $Z_f := \int_{\mathcal{D}} \exp(-f) d\mathbf{x}$ is the normalizing constant.

**Distribution and function class** If $f$ is (strongly) convex, the distribution $\pi_f$ is said to be (strongly) *log-concave*. If $f$ is $L$-Lipschitz, the distribution $\pi_f$ is said to be *L-log-Lipschitz*. If $f$ is twice-differentiable and $\nabla^2 f \preceq LI$ (where $\preceq$ denotes the Loewner order and $I$ is the identity matrix), we say the distribution $\pi_f$ is *L-log-smooth*.

**Oracle** In this work, we investigate the model where the algorithm queries points from the domain $\mathcal{D}$ to the oracle $\mathcal{O}$. Given the potential function $f$, and a query $\mathbf{x} \in \mathcal{D}$, the 0-th order oracle answers

the function value $f(\mathbf{x})$ and the 1-st order oracle answers both $f(\mathbf{x})$ and its gradient value $\nabla f(\mathbf{x})$. We will focus on the 0-th-order oracle for the rest of this paper. Our results under 0-th-order oracles can be extended to 1-th-order oracles, as 1-th-order oracles can be obtained from 0-th-order oracles when polynomially many queries are allowed per round.

**Adaptive algorithm class**  The class of *adaptive* algorithms is formally defined as follows (Diakonikolas and Guzmán, 2019). For any dimension $d$, an adaptive algorithm A takes $f : \mathbb{R}^d \to \mathbb{R}$ and a (possibly random) initial point $\mathbf{x}^0$ and iteration number $r$ as input and returns an output $\mathbf{x}^{r+1}$, which is denoted as $\mathsf{A}[f, \mathbf{x}^0, r] = \mathbf{x}^{r+1}$. At iteration $i \in [r] := \{1, \ldots, r\}$, A performs a batch of queries

$$Q^i = \{\mathbf{x}^{i,1}, \ldots, \mathbf{x}^{i,k_i}\}, \quad \text{with } \mathbf{x}^{i,j} \in \mathcal{D}, \ j \in [k_i], \ k_i = \mathsf{poly}(d),$$

such that for any $m, n \in [k_i]$, $\mathbf{x}^{i,m}$ and $\mathbf{x}^{i,n}$ are *conditionally independent* given all existing queries $\{Q^j\}_{j \in [i-1]}$ and $\mathbf{x}^0$. Give queries set $Q^i$, the oracle returns a batch of answers: $\mathcal{O}(Q^i) = \{\mathcal{O}(\mathbf{x}^{i,1}), \ldots, \mathcal{O}(\mathbf{x}^{i,k_i})\}$.

An adaptive algorithm A is *deterministic* if in every iteration $i \in \{0, \ldots, r\}$, A operates with the form $Q^{i+1} = \mathsf{A}^i(Q^0, \mathcal{O}(Q^0), \ldots, Q^i, \mathcal{O}(Q^i))$, where $\mathsf{A}^i$ is mapping into $\mathbb{R}^{dk_{i+1}}$ with $Q^{r+1} = \mathbf{x}^{r+1}$ as output and $Q^0 = \mathbf{x}^0$ as an initial point. We denote the class of adaptive deterministic algorithms by $\mathcal{A}_{\mathrm{det}}$.

An adaptive *randomized* algorithm has the form $Q^{i+1} = \mathsf{A}^i(\xi_i, Q^0, \mathcal{O}(Q^0), \ldots, Q^i, \mathcal{O}(Q^i))$, with access to a uniform random variable on $[0, 1]$ (i.e., infinitely many random bits), where $\mathsf{A}^i$ is mapping into $\mathbb{R}^{dk_{i+1}}$. We denote the class of adaptive randomized algorithms by $\mathcal{A}_{\mathrm{rand}}$.

**Measure of the output**  Consider the joint distribution of all involved points $\{\mathbf{x} : \mathbf{x} \in Q^i, i = 0, \ldots, r + 1\}$ and the random bits $\xi_i$. Let the marginal distribution of the output $\mathbf{x}^{r+1}$ be $\rho$. We say the output to be $\varepsilon$-accurate in total variation if $\mathsf{TV}(\rho, \pi_f) := \sup_{A \subseteq \mathcal{D}} |\rho(A) - \pi_f(A)| \leq \varepsilon$.

**Initialization**  For initial point $\mathbf{x}^0 \sim \rho_0$, the initial distribution $\rho_0$ is said to be $M$-infinite Rényi warm start for $\pi$ if $\mathcal{R}_\infty(\rho_0 \| \pi) \leq M$, where $\mathcal{R}_\infty(\cdot \| \cdot)$ is the infinite Rényi divergence defined as $\mathcal{R}_\infty(\mu \| \pi) = \ln \left\| \frac{\mathrm{d}\mu}{\mathrm{d}\pi} \right\|_{L^\infty(\pi)}$ [2].

**Notion of complexity**  Given $\varepsilon > 0$, $f \in \mathcal{F}$, and some algorithm A, define the running iteration $\mathsf{T}(\mathsf{A}, f, \mathbf{x}^0, \varepsilon, \mathsf{TV})$ as the minimum number of rounds such that algorithm A outputs a solution $\mathbf{x}$ whose marginal distribution $\rho$ satisfies $\mathsf{TV}(\rho, \pi_f) \leq \varepsilon$, i.e., $\mathsf{T}(\mathsf{A}, f, \mathbf{x}^0, \varepsilon, \mathsf{TV}) = \inf \{t : \mathsf{TV}(\rho(\mathsf{A}[f, \mathbf{x}^0, t]), \pi_f) \leq \varepsilon\}$ [3]. We define the *worst case* complexity as

$$\mathsf{Comp}_{\mathsf{WC}}(\mathcal{F}, \varepsilon, \mathbf{x}^0, \mathsf{TV}) := \inf_{\mathsf{A} \in \mathcal{A}_{\mathrm{det}}} \sup_{f \in \mathcal{F}} \mathsf{T}(\mathsf{A}, f, \mathbf{x}^0, \varepsilon, \mathsf{TV}).$$

For some randomized algorithm $\mathsf{A} \in \mathcal{A}_{\mathrm{rand}}$, we define the *randomized* complexity as [4]

$$\mathsf{Comp}_{\mathsf{R}}(\mathcal{F}, \varepsilon, \mathbf{x}^0, \mathsf{TV}) := \inf_{\mathsf{A} \in \mathcal{A}_{\mathrm{rand}}} \sup_{f \in \mathcal{F}} \mathsf{T}(\mathsf{A}, f, \mathbf{x}^0, \varepsilon, \mathsf{TV}).$$

By definition, we have $\mathsf{Comp}_{\mathsf{WC}}(\mathcal{F}, \varepsilon, \mathbf{x}^0, \mathsf{TV}) \geq \mathsf{Comp}_{\mathsf{R}}(\mathcal{F}, \varepsilon, \mathbf{x}^0, \mathsf{TV})$. In the rest of this paper, we only consider the randomized complexity and we lower-bound it by considering the *distributional* complexity:

$$\mathsf{Comp}_{\mathsf{D}}(\mathcal{F}, \varepsilon, \mathbf{x}^0, \mathsf{TV}) := \sup_{F \in \Delta(\mathcal{F})} \inf_{\mathsf{A} \in \mathcal{A}_{\mathrm{rand}}} \mathbb{E}_{f \sim F} \mathsf{T}(\mathsf{A}, f, \mathbf{x}^0, \varepsilon, \mathsf{TV}),$$

where $\Delta(\mathcal{F})$ is the set of probability distributions over the class of functions $\mathcal{F}$.

---

[2] Sometimes the warm start is defined as $\left\| \frac{\mathrm{d}\rho_0}{\mathrm{d}\pi} \right\|_{L^\infty(\pi)} \leq M$ (Wu et al., 2022; Mangoubi and Vishnoi, 2023a), but we adopt the version with logarithm.

[3] We note that in sampling, the iteration complexity is determined by the output of the last iteration, which is analogous to last-iteration properties in optimizations (Abernethy et al., 2019).

[4] We note that in sampling, we cannot define the randomized complexity as the expected running iteration over mixtures of deterministic algorithms as in the case of optimization (Braun et al., 2017), since the intrinsic randomness $\xi_i$ will affect the marginal distribution of output. Furthermore, Yao's minimax principle (Arora and Barak, 2009) cannot be applied, since the different definition of randomized complexity. We acknowledge that another possible option not discussed in this paper is the "Las Vegas" algorithm, which can return "failure," as described in Altschuler and Chewi (2024).

## 3 Technical overview

We begin by reviewing existing techniques for determining query complexity in sampling and adaptive complexity in optimization. We then discuss the challenges associated with applying these techniques and describe our methods for addressing them.

### 3.1 Existing techniques

**Query complexity for sampling.** The existing techniques for showing the (query) lower bound for the sampling task fall into one of two main approaches. The first one involves reducing the task of hypothesis test to the sampling task (Chewi et al., 2022b; 2023b). To do so, a family of hardness distributions is constructed. On the one hand, the hardness distributions are well-separated in total variation such that if we can sample well from the distribution with accuracy finer than the existing gaps between them, we can identify the distribution with a lower-bounded probability. On the other hand, with a limited number of queries, the probability of correctly identifying the distribution is upper-bounded by information arguments such as Fano's lemma (Cover, 1999). This methodology has been effectively applied to establish lower bounds on query complexity with constant accuracy for log-concave distributions in one or two dimensions (Chewi et al., 2022b; 2023b). However, whether such a method can be extended to high-dimensional sampling remains unknown.

The second method is to reduce another high-dimensional task to the sampling task, such as inverse trace estimation or finding stationary points (Chewi et al., 2023a;b). Specifically, they showed that if there is a (possibly randomized) algorithm that returns a sample whose marginal distribution is close enough to the target distribution w.r.t. the total variation distance, then there exists an algorithm to solve the reduced task with a high probability over the randomness of the task instance and algorithm. On the other hand, the hardness of the reduced task is shown by sharp characterizations of the eigenvalue distribution of Wishart matrices or chaining structured functions parameterized by orthogonal vectors Chewi et al. (2023a;b). However, at the current stage, this method can only work for constant or very low accuracy ($\varepsilon = \widetilde{\Theta}(d^c)$).

**Adaptive complexity in optimization.** For establishing adaptive lower bounds for submodular optimization, existing methodologies (Balkanski and Singer, 2020; Chakrabarty et al., 2022; 2023; Li et al., 2020) utilize a common framework. At a high level, these methods involve designing a family of submodular functions, parameterized by a uniformly random partition $\mathcal{P} = (P_1, \ldots, P_{r+1})$ over the coordinates ground set $[d]$. The key property of such construction is that even after receiving responses to polynomially many queries by round $i$, any (possibly randomized) algorithm does not possess any information about the elements in $(P_{i+1}, \ldots, P_{r+1})$ with a high probability over the randomness of partition $\mathcal{P}$. As a result, the solution at round $i$ will be independent of the $(P_{i+1}, \ldots, P_{r+1})$ part of the partition. Additionally, without knowing the information of $P_{r+1}$, any algorithm cannot return a good enough solution. To hide future information of partition, they used the chain structure or onion-like structure.

Furthermore, a similar approach based on the Nemirovski function parameterized by random orthogonal vectors has also been the main tool to prove lower bounds for parallel convex optimization (Balkanski and Singer, 2018b; Bubeck et al., 2019; Diakonikolas and Guzmán, 2019; Garg et al., 2021).

### 3.2 Technical challenges and our methods

Our results for log-concave distributions leverage the same framework. We construct a family of distributions parameterized by uniformly random partitions with chain structure. A similar argument applies that any adaptive algorithm cannot learn any information about $(P_{i+1}, \ldots, P_{r+1})$ before the $i$-th round. However, there are technical challenges to apply a similar information argument as Chewi et al. (2022b; 2023b), which we summarize below.

1. The hard distributions in Chewi et al. (2022b; 2023b) are well-separated, i.e., the total variation distance between any two distributions is large enough, which benefits the argument of reduction from identification to sampling. However, the hard distributions constructed by the random partition are close w.r.t. the total variation distance.

2. On the other hand, Chewi et al. (2023b) applied Fano's lemma with the advantage of bounded information gain for every query. However, it is unclear how to find the bound of all the information gains for polynomially many queries.

As a result, the reduction from a hypothesis test no longer applies, and the implications of not knowing the partition $(P_{i+1}, \ldots, P_{r+1})$ remain unknown.

The technical contributions of this paper lie in tackling these challenges by characterizing the outputs. Specifically, we show that without knowing $(P_r, P_{r+1})$, the output cannot reach the area $\{\mathbf{x} \in \mathbb{R}^d, |\sum_{i \in P_r} \mathbf{x}_i - \sum_{i \in P_{r+1}} \mathbf{x}_i| \geq t\}$ with a reasonable threshold $t$ (Lemma 4.3). This is achieved through the concentration bound of conditional Bernoulli random variables (Theorem A.1 and Theorem A.3). Without hitting such an area, we can lower-bound the total variation distance between the marginal distribution and the target distribution.

Moreover, we extend this result to the unconstrained distributions by taking the maximum operator between the original hard distribution and a function without information about the partition. Additionally, we adapt the result to the smooth case by employing a classical smoothing technique.

# 4 LOWER BOUNDS FOR UNCONSTRAINED LOG-CONCAVE SAMPLING

In this section, we construct different hardness potentials over the whole space under different smooth and convex conditions. We first present the hardness potentials for strongly log-concave and log-smooth samplers in Section 4.1.1 and its analysis in Theorem 4.1. Due to their similarity of the hardness potentials for different settings, we present the other results in Section 4.2 while we defer some proofs to Appendix B.

## 4.1 LOWER BOUND FOR STRONGLY LOG-CONCAVE SAMPLING

**Theorem 4.1 (Lower bound for strongly log-concave and log-smooth samplers).** *Let $d$ be sufficiently large, and $c \in (0, 1)$ be a fixed uniform constant. Consider the function class $\mathcal{F}$, consisting of 1-strongly convex and 2-smooth functions. For any $\alpha = \omega(1)$ and $\varepsilon = \mathcal{O}(c^d)$, there exists $\gamma = \mathcal{O}(d^{-\alpha})$ and $\mathbf{x}^0 \sim \rho_0$ with $\rho_0$ as $\widetilde{\mathcal{O}}(d)$-infinite Rényi warm start for any $\{\pi_f\}_{f \in \mathcal{F}}$ such that $\mathsf{Comp}_\mathsf{R}(\mathcal{F}, \varepsilon, \mathbf{x}^0, \mathsf{TV}) \geq (1 - \gamma)\frac{d}{\alpha \log^3 d}$.*

The assumption that $d$ is sufficiently large is necessary for achieving a high success probability and has also been adopted in prior works on the lower bounds of parallel optimization (Balkanski and Singer, 2018a; Li et al., 2020). As we are only concerned with the order of complexity, we do not estimate the exact value of $c \in (0, 1)$. Actually, we can choose any $c$ which satisfies that $\frac{\sqrt{\frac{t}{1-t}}}{2\pi\sqrt{d}(d+2)}\left(\frac{\sqrt{t}}{2}\right)^d \geq c^d$ with a fixed $t \in (0, 1)$ depending on $d$ (see Section 4.1.2).

### 4.1.1 CONSTRUCTION OF HARDNESS POTENTIALS FOR THEOREM 4.1

Before we describe the details of hardness potentials, we first recall the properties of the smoothing operator modified by Proposition 1 in Guzmán and Nemirovski (2015). We defer the details of the proof to Appendix A.3.

**Theorem 4.2 (Smoothing operators).** *Let $f : \mathbb{R}^d \to \mathbb{R}$ be a 1-Lipschitz and convex function. There exists a convex continuously differentiable function $S_1[f](\mathbf{x}) : \mathbb{R}^d \to \mathbb{R}$ with the following properties:*

1. *$f(\mathbf{x}) \geq S_1[f](\mathbf{x}) \geq f(\mathbf{x}) - 1$ for all $\mathbf{x} \in \mathbb{R}^d$;*

2. *$\|\nabla S_1[f](\mathbf{x}) - \nabla S_1[f](\mathbf{y})\|_2 \leq \|x - y\|_2$ for all $\mathbf{x}, \mathbf{y} \in \mathbb{R}^d$;*

3. *For every $\mathbf{x} \in \mathbb{R}^d$, the restriction of $S_1[f](\cdot)$ on a small enough neighbourhood of $\mathbf{x}$ depends solely on the restriction of $f$ on the set $B_1(\mathbf{x}) = \{\mathbf{y} : \|\mathbf{y} - \mathbf{x}\| \leq 1\}$.*

*Furthermore, if $f$ reaches its minimum of 0 at $\mathbf{x}^\star = 0$, then $S_1[f]$ does likewise.*

Let $d_0 \in \mathbb{N}_+$ with $d_0 \geq \log^3 d$, and $r = d/d_0 - 2$. Consider partition with fixed-size components as $P_1 \cup P_2 \cup \cdots \cup P_{r+2} = [d]$, where $|P_1| = d_0$, $|P_1| = |P_i|$ for all $i \in [r]$. The partition is uniformly

random among all partitions with such fixed-size parts, and we denote it as $\mathcal{P}$. For any $\mathbf{x} \in \mathbb{R}^d$ let $X^i = \sum_{s \in P_i} \mathbf{x}_s$ for all $i \in [r+1]$, and we denote $X(\mathbf{x}) = (X^1, \ldots, X^{r+1})$. We define the 1-strongly log-concave hardness potential $f_{\mathcal{P}} : \mathbb{R}^d \to \mathbb{R}$ as

$$f_{\mathcal{P}}(\mathbf{x}) = S_1[g_{\mathcal{P}}](\mathbf{x}) + \frac{1}{2} \|\mathbf{x}\|^2 ,$$

where $S_1$ is the smoothing operator defined in Theorem 4.2 and $g_{\mathcal{P}} : \mathbb{R}^d \to \mathbb{R}$ is defined as

$$g_{\mathcal{P}}(\mathbf{x}) = \max\left\{ L \cdot \left( |X^1| + \sum_{i \in [r]} \max\left\{ \left|X^i - X^{i+1}\right| - t, 0 \right\} \right), \|\mathbf{x}\| - M \right\},$$

where $t = 8(M+1)\sqrt{\alpha \log d}$ and $L = 1/(4\sqrt{d})$. We will specify the parameter $M$ later.

By construction, we observe that

1. When $\|\mathbf{x}\| \geq 2M$, it must be

$$L \cdot \left( |X^1| + \sum_{i \in [r]} \max\left\{ \left|X^i - X^{i+1}\right| - t, 0 \right\} \right) \leq \|\mathbf{x}\| - M,$$

   i.e., outside the ball of radius $2M$, $g_{\mathcal{P}}$ is only determined by its right term.

2. When $\|\mathbf{x}\| \leq M$, it must be

$$L \cdot \left( |X^1| + \sum_{i \in [r]} \max\left\{ \left|X^i - X^{i+1}\right| - t, 0 \right\} \right) \geq \|\mathbf{x}\| - M,$$

   i.e., within the ball of radius $M$, $g_{\mathcal{P}}$ is only determined by its left term.

At the $i$-th iteration, we assume that no information about $(P_{i+1}, \ldots, P_r)$ is processed in advance. For any query $\mathbf{x} \in Q^i$, if $\|\mathbf{x}\| \leq 2M$, the concentration of linear functions over the Boolean slice (Theorem A.3) implies that $g_{\mathcal{P}}(\mathbf{x})$ is independent of $(P_{i+1}, \ldots, P_r)$ with high probability over $\mathcal{P}$. Similarly, if $\|\mathbf{x}\| \geq 2M$, then $g_{\mathcal{P}}(\mathbf{x}) = \|\mathbf{x}\| - M$, which also demonstrates independence from $(P_{i+1}, \ldots, P_r)$. Consequently, the solution at round $i$ will be independent of the partition segment $(P_{i+1}, \ldots, P_{r+1})$. Moreover, lacking knowledge of $(P_r, P_{r+1})$, the output cannot reach the area $S = \left\{ \mathbf{x} \in \mathbb{R}^d \mid |\sum_{i \in P_r} \mathbf{x}_i - \sum_{i \in P_{r+1}} \mathbf{x}_i| \geq t \right\}$ with a reasonable threshold $t$, leveraging the concentration bounds of conditional Bernoullis (Theorem A.3). Thus, the marginal distribution $\rho$ of $\mathbf{x}^{r+1}$ satisfies $\rho(S) = 0$ with a high probability over randomness of $\mathcal{P}$.

So far we have not considered the inference of the smoothing operator. However, actually, it will only have a constant effect since it preserves the function value and the local area within a constant tolerance. The constant change in the value of the potential function will result in only a constant alteration of the mass value, whereas the change in the local area will lead to the constant shrinking of the unreachable area.

We give the formal description in the following lemma, which is our main technical contribution.

**Lemma 4.3.** *For any randomized algorithm* A, *any* $\tau \leq r$, *and any initial point* $\mathbf{x}^0$, $X(\mathsf{A}[f_{\mathcal{P}}, \mathbf{x}^0, \tau])$ *takes a form as*

$$(x^1, \ldots, x_\tau, x_\tau, \ldots, x_\tau),$$

*up to addictive error* $\mathcal{O}(t)$ *for every coordinate with probability* $1 - d^{-\omega(1)}$ *over* $\mathcal{P}$.

*Proof.* We fixed $\tau$ and prove the following by induction for $l \in [\tau]$: With high probability, the computation path of the (deterministic) algorithm A and the queries it issues in the $l$-th round are determined by $P_1, \ldots, P_{l-1}$.

As a first step, we assume the algorithm is deterministic by fixing its random bits and choose the partition of $\mathcal{P}$ uniformly at random.

To prove the inductive claim, let $\mathcal{E}_l$ denote the event that for any query $\mathbf{x}$ issued by A in iteration $l$, the answer is in the form $S_1[g_{\mathcal{P}}^l](\mathbf{x}) + \frac{1}{2} \|\mathbf{x}\|^2$ where $g_{\mathcal{P}}^l : \mathbb{R}^d \to \mathbb{R}$ defined as:

$$g_{\mathcal{P}}^l(\mathbf{x}) = \max\left\{ L \cdot \left( |X^1| + \sum_{i \in [l-1]} \max\left\{ \left|X^i - X^{i+1}\right| - t, 0 \right\} \right), \|\mathbf{x}\| - M \right\},$$

i.e., $\mathcal{E}_l$ represents the events that $\forall \mathbf{x} \in Q^l$, $f_{\mathcal{P}}(\mathbf{x}) = S_1[g_{\mathcal{P}}](\mathbf{x}) + \frac{1}{2} \|\mathbf{x}\|^2 = S_1[g_{\mathcal{P}}^l](\mathbf{x}) + \frac{1}{2} \|\mathbf{x}\|^2$.

Since the queries in round $l$ depend only on $P_1, \ldots, P_{l-1}$, if $\mathcal{E}_l$ occurs, the entire computation path in round $l$ is determined by $P_1, \ldots, P_l$. By induction, we conclude that if all of $\mathcal{E}_1, \ldots, \mathcal{E}_l$ occur, the computation path in round $l$ is determined by $P_1, \ldots, P_l$.

Now we analyze the conditional probability $P[\mathcal{E}_l \mid \mathcal{E}_1, \ldots, \mathcal{E}_{l-1}]$. By the property 3 of Theorem 4.2, $S_1[g_{\mathcal{P}}](\mathbf{x})$ only depends on $\{g_{\mathcal{P}}(\mathbf{x}) : \mathbf{x}' \in B_1(\mathbf{x})\}$. Thus, it is sufficient to analyze the probability of the event that for a fixed query $\mathbf{x}$, any point $\mathbf{x}' \in B_1(\mathbf{x})$ satisfies that $g_{\mathcal{P}}(\mathbf{x}') = g_{\mathcal{P}}^l(\mathbf{x}')$.

**Case 1:** $\|\mathbf{x}\| \leq 2M + 1$. Given all of $\mathcal{E}_1, \ldots, \mathcal{E}_{l-1}$ occur so far, we can claim that $Q^l$ is determined by $P_1, \ldots, P_l$. Conditioned on $P_1, \ldots, P_l$, the partition of $[d] \setminus \bigcup_{i \in [l]} P_i$ is uniformly random. We consider $\{0, 1\}$-random variable $Y_j$, $j \in [d] \setminus \bigcup_{i \in [l]} P_i$. For any query $\mathbf{x}' \in B_1(\mathbf{x})$, we represent $X^i(\mathbf{x}')$ as a linear function of $Y_i$s as $X^i(\mathbf{x}') = \sum_{j \in [d] \setminus \bigcup_{i \in [l]} P_i} Y_j \mathbf{x}'_j$ such that $Y_i = 1$ if $Y_i \in P_i$ and $Y_i = 0$ otherwise. By the concentration of linear functions over the Boolean slice (Theorem A.3), and recall $t = 8(M + 1)\sqrt{\alpha \log d}$, we have

$$\mathbb{P}_{\mathcal{P}}\left[ |X^i(\mathbf{x}') - \mathbb{E}[X^i(\mathbf{x}')]| \geq \frac{t}{2} \right] \leq 2\exp\left( -\frac{t^2}{16(2M + 2)^2} \right) = 2d^{-\omega(1)}.$$

Similarly, $\mathbb{P}_{\mathcal{P}}\left[ |X^{i+1}(\mathbf{x}') - \mathbb{E}[X^{i+1}(\mathbf{x}')]| \geq \frac{t}{2} \right] \leq 2d^{-\omega(1)}$. Combining the fact that $\mathbb{E}[X^i(\mathbf{x}')] = \mathbb{E}[X^{i+1}(\mathbf{x}')]$, we have with probability at least $1 - d^{-\omega(1)}$, for any fixed $i \geq l$

$$\max\left\{ |X^i(\mathbf{x}') - X^{i+1}(\mathbf{x}')| - t, 0 \right\} = 0,$$

which implies $g_{\mathcal{P}}(\mathbf{x}') = g_{\mathcal{P}}^l(\mathbf{x}')$ with a probability at least $1 - rd^{-\omega(1)}$.

**Case 2:** $\|\mathbf{x}\| \geq 2M + 1$. For any $\mathbf{x}' \in B_1(\mathbf{x})$, we have $\|\mathbf{x}'\| \geq 2M$, which implies

$$g_{\mathcal{P}}(\mathbf{x}) = \|\mathbf{x}\| - M = g_{\mathcal{P}}^l(\mathbf{x}).$$

Combining these two cases, we have for any fixed query $\mathbf{x}$, with a probability at least $1 - rd^{-\omega(1)}$, any point $\mathbf{x}' \in B_1(\mathbf{x})$ satisfies that $g_{\mathcal{P}}(\mathbf{x}') = g_{\mathcal{P}}^l(\mathbf{x}')$.

By union bound over all queries $\mathbf{x} \in Q^l$, conditioned on that $\mathcal{E}_1, \ldots, \mathcal{E}_{l-1}$ occur, with probability at least $1 - r\mathsf{poly}(d)d^{-\omega(1)}$, $\mathcal{E}_l$ occurs. Therefore by induction,

$$\begin{aligned}
\mathbb{P}(\mathcal{E}_l) &= \mathbb{P}(\mathcal{E}_l|\mathcal{E}_1, \ldots, \mathcal{E}_{l-1})\mathbb{P}(\mathcal{E}_{l-1}|\mathcal{E}_1, \ldots, \mathcal{E}_{l-2}) \ldots \mathbb{P}(\mathcal{E}_2|\mathcal{E}_1)\mathbb{P}(\mathcal{E}_1) \\
&\geq 1 - r^2\mathsf{poly}(d)d^{-\omega(1)} = 1 - d^{-\omega(1)}.
\end{aligned}$$

This implies that with high probability, the computation path in round $l$ is determined by $P_1, \ldots, P_{l-1}$. Consequently, for all $l \in [\tau]$ a solution returned after $l - 1$ rounds is determined by $P_1, \ldots, P_{l-1}$ with high probability. By the same concentration argument, the solution is with a probability at least $1 - d^{-\omega(1)}$ in the form

$$(x_1, \ldots, x_\tau, x_\tau, \ldots, x_\tau),$$

up to an additive error $\mathcal{O}(t)$ in each coordinate.

Finally, we note that by allowing the algorithm to use random bits, the results are a convex combination of the bounds above, so the same high-probability bounds are satisfied. $\qquad\square$

### 4.1.2 PROOF OF THEOREM 4.1

**Verification of $f_{\mathcal{P}}$.** Since $g_{\mathcal{P}}$ is 1-Lipschitz and convex, by Theorem 4.2, $S_1[g_{\mathcal{P}}]$ is 1-smooth and convex, which implies $f_{\mathcal{P}}(\mathbf{x}) = S_1[g_{\mathcal{P}}](\mathbf{x}) + \frac{1}{2}\|\mathbf{x}\|^2$ is 1-strongly convex and 2-smooth.

**Bound of total variation distance.** We first estimate the normalizing constant as follows.

$$\begin{aligned}
Z_{f_{\mathcal{P}}} &\leq \int_{\mathbb{R}^d} \exp\left( -g_{\mathcal{P}}(\mathbf{x}) + 1 - \frac{1}{2}\|\mathbf{x}\|^2 \right) d\mathbf{x} && \text{(by Property 1 of Theorem 4.2)} \\
&\leq e \int_{\mathbb{R}^d} \exp\left( -\|\mathbf{x}\| + M - \frac{1}{2}\|\mathbf{x}\|^2 \right) d\mathbf{x} && \text{(by definition of } g_{\mathcal{P}}) \\
&= \exp(M + 1)\frac{2\Gamma^d(1/2)}{\Gamma(d/2)} \int_0^\infty t^{d-1}\exp(-t^2/2)dt = \exp(M + 1) \cdot (2\pi)^{d/2}. && (1)
\end{aligned}$$

Consider a subset $S = \{\mathbf{x} \in \mathbb{R}^d : \|\mathbf{x}\| \leq M, X_{r+1} - X_r \geq t\}$. By Lemma 4.3, with probability $1 - d^{-\omega(1)}$ over $\mathcal{P}$, $\rho(\mathsf{A}[f_{\mathcal{P}}, \mathbf{x}^0, r])(S) = 0$. Also by Property 1 in Theorem 4.2, and definition of $f_{\mathcal{P}}$ and $g_{\mathcal{P}}$, we have for any $\mathbf{x} \in S$, and $M \geq 1$,

$$f_{\mathcal{P}}(\mathbf{x}) \leq g_{\mathcal{P}}(\mathbf{x}) + \frac{1}{2}\|\mathbf{x}\|^2 \leq \frac{1}{2}\|\mathbf{x}\| + \frac{1}{2}\|\mathbf{x}\|^2 \leq \frac{M + M^2}{2} \leq M^2.$$

Thus, we have $\pi_{f_{\mathcal{P}}}(S) = \frac{\int_S \exp(f_{\mathcal{P}}(\mathbf{x}))\mathrm{d}\mathbf{x}}{Z_{f_{\mathcal{P}}}} \geq \frac{\int_S \exp(-M^2)\mathrm{d}\mathbf{x}}{Z_{f_{\mathcal{P}}}} = \frac{|S|\exp(-M^2)}{Z_{f_{\mathcal{P}}}}$. Recall $t = 8(M + 1)\sqrt{\alpha \log d}$, we define the height of the sphere cap as $h = M - \frac{t}{\sqrt{2d_0}} = M - 8(M+1)\sqrt{\frac{\alpha \log d}{2d_0}}$. Let $b = (2Mh - h^2)/M^2 = 1 - \left(1 + \frac{1}{M}\right)^2 \frac{64\alpha \log d}{d_0}$. By Lemma A.4 and Eq. equation 1, we have

$$\pi_{f_{\mathcal{P}}}(S) \geq \frac{|S|\exp(-M^2)}{Z_{f_{\mathcal{P}}}} \geq \frac{|S|}{V_d M^d} \cdot V_d \cdot \frac{1}{(2\pi)^{d/2}} \cdot M^d \exp(-2M^2)$$

$$= \frac{I_b\left(\frac{d+1}{2}, \frac{1}{2}\right)}{2} \cdot \frac{\pi^{d/2}}{\Gamma(d/2 + 1)} \cdot \frac{1}{(2\pi)^{d/2}} \cdot \left(\frac{d}{4}\right)^{d/2} \exp\left(-\frac{d}{2}\right),$$

by taking $M^2 = d/4$, where $\Gamma(\cdot)$ is Gamma function, $I_x(a, b)$ is incomplete Beta function.

We recall $d_0 = \log^3 d$ and let $d_0 = \omega(\alpha \log d)$. For sufficient large $d$, we have $b \to 1$, which implies $b \geq t$ for any fixed $t \in (0, 1)$, $I_b\left(\frac{d+1}{2}, \frac{1}{2}\right) \geq I_t\left(\frac{d+1}{2}, \frac{1}{2}\right)$. By the expansion of the incomplete Beta function,

$$I_t\left(\frac{d+1}{2}, \frac{1}{2}\right) = \Gamma\left(\frac{d}{2} + 1\right) t^{(d+1)/2}(1 - t)^{-1/2}\left(\frac{1}{\Gamma\left(\frac{d+1}{2} + 1\right)\Gamma\left(\frac{1}{2}\right)} + \mathcal{O}\left(\frac{1}{\Gamma\left(\frac{d+1}{2} + 1\right)}\right)\right),$$

we have $I_b\left(\frac{d+1}{2}, \frac{1}{2}\right) \geq \sqrt{\frac{t}{1-t}} \cdot \frac{2}{\sqrt{\pi}} \cdot \frac{t^{d/2}}{d+2}$. Furthermore, by Stirling's approximation, $\Gamma\left(\frac{d}{2} + 1\right) = \sqrt{2\pi(d/2)}\left(\frac{d}{2e}\right)^{d/2}\left(1 + \mathcal{O}\left(\frac{1}{d}\right)\right)$, we have

$$\pi_{f_{\mathcal{P}}}(S) \geq \frac{I_b\left(\frac{d+1}{2}, \frac{1}{2}\right)}{2} \cdot \frac{1}{\Gamma(d/2 + 1)} \cdot \frac{1}{2^{d/2}} \cdot \left(\frac{d}{4}\right)^{d/2} \exp\left(-\frac{d}{2}\right) \geq \frac{\sqrt{\frac{t}{1-t}}}{2\pi\sqrt{d}(d+2)}\left(\frac{\sqrt{t}}{2}\right)^d.$$

Thus, there exists a constant $c \in (0, \frac{1}{2})$ such that with probability $1 - d^{-\omega(1)}$ over $\mathcal{P}$,

$$\mathsf{TV}(\rho(\mathsf{A}[f_{\mathcal{P}}, \mathbf{x}^0, r]), \pi_{f_{\mathcal{P}}}) \geq \Omega(c^d).$$

**Initial condition.** For any potential function $f \in \mathcal{F}$, it holds that $f$ reaches its minimum value of $0$ at the origin. Consider initial distribution $\rho_0$ as $\mathsf{normal}(0, I_d)$. By Lemma A.5 and Lemma A.6, the initial condition holds.

## 4.2 LOWER BOUNDS FOR WEAKLY LOG-CONCAVE SAMPLING

Here, we consider the potentials without strong convexity or taking composite forms. Similarly, we show exponentially small lower bounds for almost linear iterations (Theorem 4.4 and Theorem 4.5).

**Theorem 4.4 (Lower bound for weakly log-concave samplers).** *Let $d$ be sufficiently large, and $c \in (0, 1)$ be a fixed uniform constant. Consider the function class $\mathcal{F}$, consisting of convex and 1-smooth or 1-Lipschitz functions. For any $\alpha = \omega(1)$ and $\varepsilon = \mathcal{O}(c^d)$, there exists $\gamma = \mathcal{O}(d^{-\alpha})$ and $\mathbf{x}^0 \sim \rho_0$ with $\rho_0$ as $\widetilde{\mathcal{O}}(d)$-infinite Rényi warm start for any $f \in \mathcal{F}$ such that $\mathsf{Comp}_{\mathsf{R}}(\mathcal{F}, \varepsilon, \mathbf{x}^0, \mathsf{TV}) \geq (1 - \gamma)\frac{d}{\alpha \log^3 d}$.*

The proof can be found in Appendix B. The main difference from Theorem 4.1 is the requirement of the initial point. Since any strongly log-concave distribution always has a bounded second moment, $\rho_0 \sim \mathsf{normal}(0, I_d)$ can be $\widetilde{\mathcal{O}}(d)$-warm start. However, a weakly log-concave distribution may have an unbounded second moment, leading to $\rho_0 \sim \mathsf{normal}(0, I_d)$ not being a warm start, and our lower bound uncomparable to the existing upper bound. As for the smoothness condition, the smoothing operator only causes a constant change in function value and a constant expansion of the local structure, which will not change the order of the lower bound. We note that changing from a strongly log-concave distribution to a weakly log-concave one will only increase the logarithmic base $c$ without changing the complexity order for our constructions (Eq. equation 5). However, the lower bound remains almost exponential because the proportion of the unreachable area is exponentially small.

### 4.3 LOWER BOUND FOR COMPOSITE SAMPLING

**Theorem 4.5 (Lower bound for composite potentials).** *Let $d$ be sufficiently large, and $c \in (0, 1)$ be a fixed uniform constant. Consider the potential function class $\mathcal{F}$, consisting of functions $f(\cdot) = f_1(\cdot) + \frac{1}{2}\|\cdot\|^2$, where $f_1$ is convex, $1$-Lipschitz. For any $\alpha = \omega(1)$ and $\varepsilon = \mathcal{O}(c^d)$, there exists $\gamma = \mathcal{O}(d^{-\alpha})$ and $\mathbf{x}^0 \sim \rho_0$ with $\rho_0$ as $\widetilde{\mathcal{O}}(d)$-warm start for any $f \in \mathcal{F}$ such that $\mathsf{Comp}_{\mathsf{R}}(\mathcal{F}, \varepsilon, \mathbf{x}^0, \mathsf{TV}) \geq (1 - \gamma)\frac{d}{\alpha \log^3 d}$.*

The proof can be found in Appendix C. It is almost the same as Theorem 4.1 except for the smoothness condition. Similarly, the smoothing operator will not change the order of the complexity.

## 5 LOWER BOUNDS FOR BOX-CONSTRAINED LOG-CONCAVE SAMPLERS

In various applications, such as Bayesian inference and differential privacy Silvapulle and Sen (2011); McSherry and Talwar (2007), the domain is constrained by polytopes. The simplest form of constraint is the box constraint, and we show the impossibility of obtaining a sup-polynomially small accuracy sample within almost linear iterations (Theorem 5.1) for box-constrained samplers.

**Theorem 5.1 (Lower bound for log-smooth and log-concave samplers).** *Let $d$ be sufficiently large, and $c \in (0, 1)$ be a fixed uniform constant. Consider the potential function class $\mathcal{F}$ as convex and $1$-smooth or $1$-Lipschitz function over the cube centered at the origin with length $1$. For any $\varepsilon = \Omega(d^{-\omega(1)})$, there exists $\gamma = \mathcal{O}(d^{-\omega(1)})$ and $\mathbf{x}^0 \sim \rho_0$ with $\rho_0$ as $\widetilde{\mathcal{O}}(d)$-infinite Rényi warm start for any $\{\pi_f\}_{f \in \mathcal{F}}$ such that $\mathsf{Comp}_{\mathsf{R}}(\mathcal{F}, \varepsilon, \mathbf{x}^0, \mathsf{TV}) \geq (1 - \gamma)\frac{d}{\log^3 d}$.*

The proof can be found in Appendix D. The main difference from Theorem 4.4 is that the lower bound holds for low-accuracy samplers. The reason is that to obscure the information, it is sufficient to construct an unreachable area with a $d^{-\omega(1)}$ proportion of the support set while keeping the distribution value bounded by a uniform constant. Similarly, taking advantage of the smoothing operator, we can prove that the smooth case is almost the same as the Lipschitz case with constant changes in distribution value and local structure.

## 6 CONCLUSIONS AND FUTURE WORKS

This work established adaptive lower bounds for log-concave distributions in various settings. We demonstrated almost linear lower bounds for unconstrained samplers with specific exponentially small accuracy for (i) strongly log-concave and log-smooth, (ii) log-concave and log-smooth or log-Lipschitz, and (iii) composite distributions. Additionally, we proved that box-constrained samplers cannot achieve sup-polynomially small accuracy within almost linear iterations. Our adaptive lower bounds also introduced new lower bounds for query complexity. Our proof relies upon novel analysis with the characterization of the output for the hardness potentials based on the chain-like structure with random partition and classical smoothing techniques.

However, these bounds are applicable only in high-dimensional settings. In low-dimensional settings, even the query complexity of high-accuracy samplers remains unclear. Furthermore, it is not known whether our bounds are tight in all settings. Therefore, it is an open question to design optimal algorithms or find optimal lower bounds.

Furthermore, extending our lower bounds to related tasks, such as sampling for diffusion models, presents an interesting direction for future work. While there has been considerable progress on parallel sampling methods for diffusion models (Shih et al., 2024; Gupta et al., 2024; Chen et al., 2024), the theoretical lower bounds for these methods remain unexplored. Notably, sampling methods in diffusion models focus on simulating a reverse dynamics, while log-concave sampling is not solely grounded in some dynamics. This distinction introduces unique challenges for deriving lower bounds for diffusion models.

## ACKNOWLEDGMENTS

The authors thank Sinho Chewi for very helpful conversations. HZ was supported by International Graduate Program of Innovation for Intelligent World and Next Generation Artificial Intelligence Research Center. BW was partially supported by the National Natural Science Foundation of China (62106213, 72394361) and an extended support project from the Shenzhen Science and Technology Program. MS was supported by the Institute for AI and Beyond, UTokyo and by a grant from Apple, Inc. Any views, opinions, findings, and conclusions or recommendations expressed in this material are those of the authors and should not be interpreted as reflecting the views, policies or position, either expressed or implied, of Apple Inc.

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

# A USEFUL TOOLS

## A.1 CONCENTRATION AND TAIL BOUNDS

**Lemma A.1 (Concentration of Lipshictz function of conditioned Bernoullis).** *Let $X_1, \ldots, X_n$ be $\{0,1\}$ random variables conditioned on $\sum\limits_{i=1}^{n} X_i = k$. Let $f : \{0,1\}^n \to \mathbb{R}$ be a 1-Lipschitz function[5]. Then for any $t > 0$,*

$$Pr[|f(X_1, \ldots, X_n) - \mathbb{E}[f(X_1, \ldots, X_n)]| \geq t] \leq 2\exp\left(-\frac{t^2}{8k}\right).$$

**Lemma A.2 (Irwin-Hall tail bound (Corollary 5 in Zhang and Zhou (2020))).** *Suppose $Y$ follows the Irwin-Hall distribution with parameter $k$, i.e., $Y = \sum\limits_{i=1}^{k} U_i$ where $U_i \sim \mathcal{U}[0,1]$. Denote $X = Y - \frac{k}{2}$. Then for $0 < x < k/2$,*

$$\mathbb{P}[X \leq -x] = \mathbb{P}[X \geq x] \leq \exp\left(-\frac{2x^2}{k}\right).$$

*There also exists constants $0 < c_0 < 1$, such that for all $\sqrt{k} \leq x \leq \frac{3k}{400}$, we have*

$$\mathbb{P}[X \leq -x] = \mathbb{P}[X \geq x] \geq c_0 \cdot \exp\left(-978\frac{x^2}{k}\right).$$

**Theorem A.3 (Concentration of linear functions over the Boolean slice (Theorem 4.2.5 in Polaczyk (2023))).** *Let $X_1, \ldots, X_n$ be $\{0,1\}$ random variables conditioned on $\sum\limits_{i=1}^{n} X_i = k$. Let $f : \{0,1\}^n \to \mathbb{R}$ be $f(\mathbf{x}) = \sum\limits_{i=1}^{n} \alpha_i \mathbf{x}_i$ with $\alpha_i \geq 0$ for all $i \in [n]$. Then for any $t > 0$,*

$$\mathbb{P}[|f(X_1, \ldots, X_n) - \mathbb{E}[f(X_1, \ldots, X_n)]| \geq t] \leq 2\exp\left(-\frac{t^2}{16\sum\limits_{i=1}^{k}(\alpha_i^{\downarrow})^2}\right),$$

*where for a finite sequence $x$, we denote by $x^{\downarrow}$ the non-increasing rearrangement of the elements of $x$.*

**Theorem A.4 (Volume of cap Li (2010)).** *The volume of cap $V \subseteq \mathbb{R}^d$ with height $h$ and radius $r$ is given by*

$$V = \frac{1}{2}V_d r^n I_{(2rh-h^2)/r^2}\left(\frac{d+1}{2}, \frac{1}{2}\right).$$

*where $I_x(a, b)$ is the regularized incomplete beta function, $V_d$ is the volume of d-dimensional ball.*

## A.2 INITIALIZATION

**Lemma A.5 (Initialization (Chewi et al., 2022a, Lemma 29)).** *Suppose that $f$ is convex with $f(0) = 0$ and $\nabla f(0) = 0$, and assume that $\nabla f$ is L-Lipschitz. Consider distribution $\pi \propto \exp(-f)$. Let $\mathsf{m} := \int_{\mathbb{R}} \|\cdot\| \, \mathrm{d}\pi$. Then, for $\mu_0 = \mathsf{normal}(0, L^{-1}I_d)$,*

$$\mathcal{R}_\infty(\mu_0 \| \pi) \leq 2 + \frac{d}{2}\ln(\mathsf{m}^2 L),$$

*where $\mathcal{R}_q(\mu\|\pi)$ is Renýi divergence defined as, for $q \in (1, \infty)$, $R_q(\mu\|\pi) = \frac{1}{q-1}\ln\left\|\frac{\mathrm{d}\mu}{\mathrm{d}\pi}\right\|_{L^q(\pi)}^q$.*

Rényi divergence is monotonic in the order: if $1 < q \leq q'$, then $\mathcal{R}_q \leq \mathcal{R}_{q'}$. We also note that if $q \to 1$, it is identical to the KL divergence, $H_\pi(\rho)$, and if $q = 2$, it is related to the chi-squared divergence via $\mathcal{R}_2(\rho\|\pi) = \ln\left(1 + \chi_\pi^2(\rho)\right)$.

---

[5] A function $f : 0, 1^n \to \mathbb{R}$ is $c$-Lipschitz, if each variable can affect the value additively by at most $c$.

**Lemma A.6** (**Bounded second moment for strongly log-concave distributions, (Dalalyan et al., 2022, Proposition 2)**). *Suppose $\pi \propto \exp(-f)$ is $\alpha$-strongly log-concave with mode $\mathbf{x}^\star$, then it holds that $\int \|\cdot - \mathbf{x}^\star\|_2^2 \, d\pi \leq \frac{d}{\alpha}$.*

### A.3 SMOOTHING APPROXIMATION

We briefly recall the results shown in Guzmán and Nemirovski (2015). For $\chi > 0$ and 1-Lipschitz continuous and convex function $f : E \to \mathbb{R}$, let

$$\mathcal{S}_\chi[f](x) = \min_{h \in \chi \mathsf{Dom}\phi} [f(x) + \chi\phi(h/\chi)],$$

where $\phi$ is smoothing kernel which is a twice continuously differentiable convex function defined on an open convex set $\mathsf{Dom}\phi \subseteq E$ with the following properties,

1. $0 \in \mathsf{Dom}\phi$ and $\phi(0) = 0$, $\phi'(0) = 0$;
2. There exists a compact convex set $G \subseteq \mathsf{Dom}\phi$ such that $0 \in \mathrm{int}G$ and $\phi(x) > \|x\|$ for all $x \in \partial G$.
3. For some $M_\phi < \infty$ we have $\forall e \in E, h \in G$,

$$\langle e, \nabla^2\phi(h)e_i \rangle \leq M_\phi \|e\|^2.$$

Then the smoothing approximation $\mathcal{S}_\chi[f](x)$ satisfies

1. $\mathcal{S}_\chi[f](x)$ is convex and Lipschitz continuous with constant 1 w.r.t. $\|\cdot\|$ and has a Lipschitz continuous gradient, with constant $M_\phi/\chi$, w.r.t. $\|\cdot\|$: for any $x, y$

$$\|\nabla\mathcal{S}_\chi[f](x) - \mathcal{S}_\chi[f](y)\|_* \leq \chi^{-1}M_\phi \|x - y\|.$$

2. $\sup_{x \in E} |f(x) - \mathcal{S}_\chi[f](x)| \leq \chi\rho_{\|\cdot\|}(G)$, where $\rho_{\|\cdot\|}(G) = \max_{h \in G} \|h\|$. Moreover, $f(x) \geq \mathcal{S}_\chi[f](x) \geq f(x) - \chi\rho_{\|\cdot\|}(G)$.

3. $\mathcal{S}_\chi[f]$ depends on f in a local fashion: the value and the derivative of $\mathcal{S}_\chi[f]$ at $x$ depends only on the restriction of $f$ onto the set $x + \chi G$.

If we choose $\chi = 1$, $\|\cdot\| = \|\cdot\|_2$, $\phi(x) = 2\|x\|_2^2$ with $M_\phi = 1$ and $G = \{x : \|x\| \leq 1\}$, we obtain properties 1-3 in Theorem 4.2.

Finally we show the minimum point will not change. By the definition of $S_1[f]$,

$$S[f](x) = f(x + h(x)) + \phi(h(x)),$$

where $h : E \to G$ is well defined and solves the nonlinear system of equations

$$F(x, h(x)) = 0, F(x, h) := f'(x + h) + \phi'(h).$$

Also, we have

$$\nabla S[f](x) = -\phi'(h(x)).$$

Thus

$$\nabla S[f](x) = 0 \Rightarrow h(x) = 0 \Rightarrow F(x, h) = F(x, 0) = f'(x) + \phi'(0) = f'(x) = 0 \Rightarrow x = 0.$$

Furthermore, when $x = 0$, we have

$$F(0, h) = f'(h) + \phi'(h) = 0,$$

which implies $h = 0$, Thus $S[f](0) = f(0 + h(0)) + \phi(h(0)) = 0$.

## B PROOF OF THEOREM 4.4

### B.1 PROOF OF SMOOTH CASE

We consider the hardness functions $f_\mathcal{P} : \mathbb{R}^d \to \mathbb{R}$:

$$f_\mathcal{P}(\mathbf{x}) = S_1[g_\mathcal{P}](\mathbf{x}),$$

where $S_1$ and $g_{\mathcal{P}}$ are as defined in Section 4.1.1, but we allow $L$ to be much smaller, such that

$$L \cdot \left( |X^1| + \sum_{i \in [r]} \max \left\{ |X^i - X^{i+1}| - t, 0 \right\} \right) \leq \Delta \text{ for all } \mathbf{x} \in B_{2M}.$$

Similarly, we have the following characterization of the output.

**Lemma B.1.** *For any randomized algorithm* $\mathsf{A}$, *any* $\tau \leq r$, *and any initial point* $\mathbf{x}^0$, $X(\mathsf{A}[f_{\mathcal{P}}, \mathbf{x}^0, \tau])$ *takes form as*

$$(x_1, \ldots, x_\tau, x_\tau, \ldots, x_\tau),$$

*up to addictive error* $\mathcal{O}(t/2)$ *with probability* $1 - d^{-\omega(1)}$ *over* $\mathcal{P}$.

We omit the proof of Lemma B.1 since it is almost the same as the proof of Lemma 4.3.

Now we are ready to prove the smooth case of Theorem 4.4.

**Verification of** $f_{\mathcal{P}}$. Since $g_{\mathcal{P}}$ is 1-Lipschitz and convex, by Theorem 4.2, $S_1[g_{\mathcal{P}}]$ is 1-smooth and convex.

**Bound of total variation distance.** We first estimate the normalizing constant as follows.

$$
\begin{aligned}
Z_{f_{\mathcal{P}}} &= \int_{\mathbb{R}^d} \exp\left(-S_\delta[g_{\mathcal{P}}](\mathbf{x})\right) \mathrm{d}\mathbf{x} \\
&\leq \int_{\mathbb{R}^d} \exp\left(-g_{\mathcal{P}}(\mathbf{x}) + 1\right) \mathrm{d}\mathbf{x} && \text{(by Property 1 of Theorem 4.2)} \\
&\leq e \int_{\mathbb{R}^d \setminus B_M} \exp\left(-\|\mathbf{x}\| + M\right) \mathrm{d}\mathbf{x} + e \int_{B_M} \mathrm{d}\mathbf{x} && \text{(by definition of } g_{\mathcal{P}}\text{)} \\
&= \exp(M+1) \frac{2\Gamma^d(1/2)}{\Gamma(d/2)} \int_M^\infty t^{d-1} \exp(-t) \mathrm{d}t + \frac{2\Gamma^d(1/2)}{\Gamma(d/2)} \int_0^M t^{d-1} \mathrm{d}t. && (2)
\end{aligned}
$$

Consider a subset $S = \left\{ \mathbf{x} \in \mathbb{R}^d : \|\mathbf{x}\| \leq M, X^{r+1} - X^r \geq t \right\}$. By Lemma B.1, with probability $1 - d^{-\omega(1)}$ over $\mathcal{P}$,

$$\rho(\mathsf{A}[f_{\mathcal{P}}, \mathbf{x}^0, r])(S) = 0.$$

Also, by Property 1 in Theorem 4.2, and definition of $f_{\mathcal{P}}$ and $g_{\mathcal{P}}$, we have for any $\mathbf{x} \in S$,

$$f_{\mathcal{P}}(\mathbf{x}) \leq g_{\mathcal{P}}(\mathbf{x}) \leq \Delta.$$

Thus, we have

$$\pi_{f_{\mathcal{P}}}(S) = \frac{\int_S \exp(f_{\mathcal{P}}(\mathbf{x})) \mathrm{d}\mathbf{x}}{Z_{f_{\mathcal{P}}}} \geq \frac{\int_S \exp(-\Delta) \mathrm{d}\mathbf{x}}{Z_{f_{\mathcal{P}}}} = \frac{|S| \exp(-\Delta)}{Z_{f_{\mathcal{P}}}}.$$

Recall $t = 8(M+1)\sqrt{\alpha \log d}$, we define the height of the sphere cap as

$$h = M - \frac{t}{\sqrt{2d_0}} = M - 8(M+1)\sqrt{\frac{\alpha \log d}{2d_0}}.$$

Let $b = (2Mh - h^2)/M^2 = 1 - \left(1 + \frac{1}{M}\right)^2 \frac{64\alpha \log d}{d_0}$. By Lemma A.4 and Eq. equation 2, we have

$$
\begin{aligned}
\pi_{f_{\mathcal{P}}}(S) &\geq \frac{|S| \exp(-\Delta)}{Z_{f_{\mathcal{P}}}} \\
&\geq \frac{|S|}{V_d M^d} \cdot \frac{\frac{2\Gamma^d(1/2)}{\Gamma(d/2)} \int_0^M t^{d-1} \mathrm{d}t}{\exp(M+1)\frac{2\Gamma^d(1/2)}{\Gamma(d/2)} \int_M^\infty t^{d-1} \exp(-t) \mathrm{d}t + \frac{2\Gamma^d(1/2)}{\Gamma(d/2)} \int_0^M t^{d-1} \mathrm{d}t} \cdot \exp(-\Delta) \\
&= \frac{I_b\left(\frac{d+1}{2}, \frac{1}{2}\right)}{2} \cdot \frac{\int_0^M t^{d-1} \mathrm{d}t}{\exp(M+1) \int_M^\infty t^{d-1} \exp(-t) \mathrm{d}t + \int_0^M t^{d-1} \mathrm{d}t} \cdot \exp(-\Delta) \\
&= \frac{I_b\left(\frac{d+1}{2}, \frac{1}{2}\right)}{2} \cdot \frac{\int_0^{d-1} t^{d-1} \mathrm{d}t}{\exp(d) \int_{d-1}^\infty t^{d-1} \exp(-t) \mathrm{d}t + \int_0^{d-1} t^{d-1} \mathrm{d}t} \cdot \exp(-\Delta), && (3)
\end{aligned}
$$

by taking $M = d - 1$.

We first estimate $\int_{d-1}^{\infty} t^{d-1} \exp(-t)\mathrm{d}t$ as

$$\lim_{d \to \infty} \frac{\int_{d-1}^{\infty} t^{d-1} \exp(-t)\mathrm{d}t}{\Gamma(d)} = \lim_{d \to \infty} \frac{\Gamma(d, d-1)}{\Gamma(d)} = \frac{1}{2}.$$

Thus for sufficiently large $d$, we have $\int_{d-1}^{\infty} t^{d-1} \exp(-t)\mathrm{d}t \le 2\Gamma(d-1)$, which implies that

$$\frac{\int_0^{d-1} t^{d-1}\mathrm{d}t}{\exp(d) \int_{d-1}^{\infty} t^{d-1} \exp(-t)\mathrm{d}t + \int_0^{d-1} t^{d-1}\mathrm{d}t} \ge \frac{(d-1)^d}{2\exp(d)\Gamma(d+1) + (d-1)^d}.$$

Furthermore, by Stirling's approximation, $\Gamma(d+1) = \sqrt{2\pi d}\left(\frac{d}{e}\right)^d \left(1 + O\left(\frac{1}{d}\right)\right)$, we have

$$\begin{aligned}
\frac{\int_0^{d-1} t^{d-1}\mathrm{d}t}{\exp(d) \int_{d-1}^{\infty} t^{d-1} \exp(-t)\mathrm{d}t + \int_0^{d-1} t^{d-1}\mathrm{d}t} &\ge \frac{(d-1)^d}{2\exp(d)d\Gamma(d+1) + (d-1)^d} \\
&\ge \frac{(d-1)^d}{4\exp(d)\sqrt{2\pi d}\left(\frac{d}{e}\right)^d + (d-1)^d} \\
&= \frac{(d-1)^d}{4\sqrt{2\pi d} \cdot d^d + (d-1)^d} \\
&= \frac{(1-1/d)^d}{4\sqrt{2\pi d} + (1-1/d)^d} \\
&\ge \frac{1}{8e\sqrt{2\pi d}}
\end{aligned} \tag{4}$$

We also recall $d_0 = \log^3 d$ and let $d_0 = \omega(\alpha \log d)$. For sufficient large $d$, for any fixed $t \in (0,1)$, we have $I_b\left(\frac{d+1}{2}, \frac{1}{2}\right) \ge \sqrt{\frac{t}{1-t}} \cdot \frac{2}{\sqrt{\pi}} \cdot \frac{t^{d/2}}{d+2}$. Combining Eq. equation 3, equation 4, we have

$$\begin{aligned}
\pi_{f_{\mathcal{P}}}(S) &\ge \sqrt{\frac{t}{1-t}} \cdot \frac{2}{\sqrt{\pi}} \cdot \frac{t^{d/2}}{d+2} \cdot \frac{1}{8e\sqrt{2\pi d}} \exp(-\Delta) \\
&\ge \frac{t^{d/2}}{4e\pi\sqrt{2}\exp(\Delta)(d+2)}.
\end{aligned} \tag{5}$$

Thus, there exists a constant $c$ such that with probability $1 - d^{-\omega(1)}$ over $\mathcal{P}$,

$$\mathsf{TV}(\rho(\mathsf{A}[f_{\mathcal{P}}, \mathbf{x}^0, r]), \pi_{f_{\mathcal{P}}}) \ge \Omega(c^d).$$

**Initial condition** Finally, we estimate the upper bound of the second moment of $\pi_{f_{\mathcal{P}}}$ for any $\mathcal{P}$,

$$\begin{aligned}
\mathsf{m}_2 &= \int_{\mathbb{R}^d} \|\mathbf{x}\|^2 \pi_{f_{\mathcal{P}}}(\mathbf{x})\mathrm{d}\mathbf{x} \\
&= \frac{1}{Z_{f_{\mathcal{P}}}} \int_{\mathbb{R}^d} \|\mathbf{x}\|^2 \exp(-f_{\mathcal{P}}(\mathbf{x}))\mathrm{d}\mathbf{x}.
\end{aligned}$$

We first upper bound the integral as

$$\begin{aligned}
\int_{\mathbb{R}^d} \|\mathbf{x}\|^2 \exp(-f_{\mathcal{P}}(\mathbf{x}))\mathrm{d}\mathbf{x} &\le \int_{\mathbb{R}^d} \|\mathbf{x}\|^2 \exp(-g_{\mathcal{P}}(\mathbf{x}) + 1)\mathrm{d}\mathbf{x} \quad \text{(By Property 1 of Theorem 4.2)} \\
&\le e \int_{\mathbb{R}^d \setminus B_M} \|\mathbf{x}\|^2 \exp(-\|\mathbf{x}\| + M)\mathrm{d}\mathbf{x} + e \int_{B_M} \|\mathbf{x}\|^2 \mathrm{d}\mathbf{x} \\
&= e \frac{2\Gamma^d(1/2)}{\Gamma(d/2)} \int_M^{\infty} t^{d-1} t^2 \exp(-t + M)\mathrm{d}t + e\frac{2\Gamma^d(1/2)}{\Gamma(d/2)} \int_0^M t^{d-1} t^2 \mathrm{d}t
\end{aligned}$$

The second inequality is implied from $g_{\mathcal{P}}(\mathbf{x}) \ge \max\{\|\mathbf{x}\| - M, 0\}$.

Then we lower bound the normalization constant as

$$
\begin{aligned}
Z_{f_{\mathcal{P}}} &= \int_{\mathbb{R}^d} \exp(-f_{\mathcal{P}}(\mathbf{x})) \mathrm{d}\mathbf{x} \\
&\geq \int_{\mathbb{R}^d} \exp(-g_{\mathcal{P}}(\mathbf{x})) \mathrm{d}\mathbf{x} && \text{(By Property 1 of Theorem 4.2)} \\
&\geq \int_{\mathbb{R}^d \setminus B_{M+\Delta}} \exp(-\|\mathbf{x}\| + M) \mathrm{d}\mathbf{x} + \int_{B_{M+\Delta}} \exp(-\Delta) \mathrm{d}\mathbf{x} \\
&= \frac{2\Gamma^d(1/2)}{\Gamma(d/2)} \int_{M+\Delta}^{\infty} t^{d-1} \exp(-t + M) \mathrm{d}t + \frac{2\Gamma^d(1/2)}{\Gamma(d/2)} \int_0^{M+\Delta} t^{d-1} \exp(-\Delta) \mathrm{d}t.
\end{aligned}
$$

The second inequality is implied from $g_{\mathcal{P}}(\mathbf{x}) \leq \max\{\|\mathbf{x}\| - M, \Delta\}$, with sufficiently small $L$ such that $L \cdot \left( |X^1| + \sum_{i \in [r]} \max\{|X^i - X^{i+1}| - t, 0\} \right) \leq \Delta$ for all $\mathbf{x} \in B_{2M}$.

Now the goal is to bound

$$
\frac{\int_M^{\infty} t^{d+1} \exp(-t + M) \mathrm{d}t + \int_0^M t^{d+1} \mathrm{d}t}{\int_{M+\Delta}^{\infty} t^{d-1} \exp(-t + M) \mathrm{d}t + \int_0^{M+\Delta} t^{d-1} \exp(-\Delta) \mathrm{d}t}.
$$

For the first term in the numerator, we have

$$
\begin{aligned}
&\int_M^{\infty} t^{d+1} \exp(-t + M) \mathrm{d}t \\
&= e^M \int_M^{\infty} t^{d+1} \exp(-t) \mathrm{d}t \\
&= e^{d/2} \Gamma\left(d+1, \frac{d}{2}\right) && \text{(By } M = \tfrac{d}{2}) \\
&= e^{d/2} d! e^{-d/2} e_d\left(\frac{d}{2}\right) && \text{(By } \Gamma(n+1, z) = n! e^{-z} e_n(z)) \\
&= d! e_d\left(\frac{d}{2}\right) \\
&\leq 2 d! e^{d/2} && \text{(By } e_d\left(\tfrac{d}{2}\right) \sim e^{d/2})
\end{aligned}
$$

where $e_n(x) = \sum_{k=0}^{n} \frac{x^k}{k!}$ is the truncated Taylor series for the exponential function, $M = \frac{d}{2}$, and $d$ is sufficiently large. For the second term in the numerator, by Stirling's approximation, $d! \sim \sqrt{2\pi d}\left(\frac{d}{e}\right)^d$, we have

$$
\int_0^M t^{d-1} t^2 \mathrm{d}t = \frac{1}{d+2}\left(\frac{d}{2}\right)^{d+2} = \mathcal{O}(d! e^{d/2}).
$$

For the first term in the denominator, we have

$$
\begin{aligned}
&\int_{M+\Delta}^{\infty} t^{d-1} \exp(-t + M) \mathrm{d}t \\
&= e^M \int_{M+\Delta}^{\infty} t^{d-1} \exp(-t) \mathrm{d}t \\
&= e^{d/2} \Gamma\left(d-1, \frac{d}{2} + \Delta\right) && \text{(By } M = \tfrac{d}{2}) \\
&= e^{d/2} (d-2)! e^{-d/2 - \Delta} e_{d-2}\left(\frac{d}{2} + \Delta\right) && \text{(By } \Gamma(n+1, z) = n! e^{-z} e_n(z)) \\
&= e^{-\Delta} (d-2)! e_{d-2}\left(\frac{d}{2} + \Delta\right) \\
&\geq \frac{e^{-\Delta} (d-2)! e^{d/2} (1 - o(1))}{2} && \text{(By } e_d\left(\tfrac{d}{2}\right) \sim e^{d/2})
\end{aligned}
$$

Thus

$$\lim_{d \to \infty} \mathbb{E}_{\pi_{f_{\mathcal{P}}}} \left[ \|\mathbf{x}\|^2 \right] \leq 4e \frac{d! e^{d/2}(1 + \mathcal{O}(1))}{e^{-\Delta}(d-2)! e^{d/2}(1 - o(1))} = 4e^{\Delta+1} d(d+1).$$

By Lemma A.5, we have the initialization condition of Renýi divergence as,

$$R_{\infty}(\mu_0 \| \pi_{f_{\mathcal{P}}}) = \tilde{\mathcal{O}}(d).$$

## B.2 PROOF OF LIPSCHITZ CASE

We consider the hardness functions $f_{\mathcal{P}} : \mathbb{R}^d \to \mathbb{R}$:

$$f_{\mathcal{P}}(\mathbf{x}) = g_{\mathcal{P}}(\mathbf{x}),$$

where $g_{\mathcal{P}} : \mathbb{R}^d \to \mathbb{R}$ is defined as Appendix C.

Similarly, we have the following characterization of the output.

**Lemma B.2.** *For any randomized algorithm* A, *any* $\tau \leq r$, *and any initial point* $\mathbf{x}^0$, $X(A[f_{\mathcal{P}}, \mathbf{x}^0, \tau])$ *takes form as*

$$(x_1, \ldots, x_\tau, x_\tau, \ldots, x_\tau),$$

*up to addictive error* $O(t/2)$ *with probability* $1 - d^{-\omega(1)}$ *over* $\mathcal{P}$.

We omit the proof of Lemma B.2 since it is almost the same as the proof of Lemma 4.3.

Now we are ready to prove the Lipschitz case of Theorem 4.4.

**Verification of** $f_{\mathcal{P}}$. It is clear that $g_{\mathcal{P}}$ is 1-Lipschitz and convex.

**Bound of total variation distance.** We omit the proof since it is almost the same as the smooth case except for scaling with constant $e$ due to the smoothing operator.

**Initial condition.** We omit the proof since it is almost the same as the smooth case except for scaling with constant $e$.

## C PROOF OF THEOREM 4.5

We consider the hardness functions $f_{\mathcal{P}} : \mathbb{R}^d \to \mathbb{R}$:

$$f_{\mathcal{P}}(\mathbf{x}) = g_{\mathcal{P}}(\mathbf{x}) + \frac{1}{2} \|\mathbf{x}\|^2,$$

where $g_{\mathcal{P}} : \mathbb{R}^d \to \mathbb{R}$ is defined as,

$$g_{\mathcal{P}}(\mathbf{x}) = \max \left\{ L \cdot \left( |X^1| + \sum_{i \in [r]} \max \left\{ |X^i - X^{i+1}| - t, 0 \right\} \right), \|\mathbf{x}\| - M \right\},$$

where $t = 8M\sqrt{\alpha \log d}$ and $L = \frac{1}{4\sqrt{d}}$.

Similarly, we have the following characterization of the output.

**Lemma C.1.** *For any randomized algorithm* A, *any* $\tau \leq r$, *and any initial point* $\mathbf{x}^0$, $X(A[f_{\mathcal{P}}, \mathbf{x}^0, \tau])$ *takes form as*

$$(x_1, \ldots, x_\tau, x_\tau, \ldots, x_\tau),$$

*up to addictive error* $\mathcal{O}(t/2)$ *with probability* $1 - d^{-\omega(1)}$ *over* $\mathcal{P}$.

*Proof of Lemma C.1.* We fixed $\tau$ and prove the following by induction for $l \in [\tau]$: With high probability, the computation path of the (deterministic) algorithm A and the queries it issues in the $l$-th round are determined by $P_1, \ldots, P_{l-1}$.

As a first step, we assume the algorithm is deterministic by fixing its random bits and choose the partition of $\mathcal{P}$ uniformly at random.

To prove the inductive claim, let $\mathcal{E}_l$ denote the event that for any query $\mathbf{x}$ issued by A in iteration $l$, the answer is in the form $g_{\mathcal{P}}^l(\mathbf{x}) + \frac{1}{2}\|\mathbf{x}\|^2$ where $g_{\mathcal{P}}^l : \mathbb{R}^d \to \mathbb{R}$ defined as:

$$g_{\mathcal{P}}^l(\mathbf{x}) = \max\left\{ L \cdot \left(|X^1| + \sum_{i \in [l-1]} \max\left\{|X^i - X^{i+1}| - t, 0\right\}\right), \|\mathbf{x}\| - M \right\},$$

i.e., $\mathcal{E}_l$ represents the events that $\forall \mathbf{x} \in Q^l$, $f_{\mathcal{P}}(\mathbf{x}) = g_{\mathcal{P}}(\mathbf{x}) + \frac{1}{2}\|\mathbf{x}\|^2 = g_{\mathcal{P}}^l(\mathbf{x}) + \frac{1}{2}\|\mathbf{x}\|^2$.

Since the queries in round $l$ depend only on $P_1, \ldots, P_{l-1}$, if $\mathcal{E}_l$ occurs, the entire computation path in round $l$ is determined by $P_1, \ldots, P_l$. By induction, we conclude that if all of $\mathcal{E}_1, \ldots, \mathcal{E}_l$ occur, the computation path in round $l$ is determined by $P_1, \ldots, P_l$.

Now we analysis the conditional probability $P\left[\mathcal{E}_l \mid \mathcal{E}_1, \ldots, \mathcal{E}_{l-1}\right]$.

**Case 1:** $\|\mathbf{x}\| \leq 2M$. Given all of $\mathcal{E}_1, \ldots, \mathcal{E}_{l-1}$ occur so far, we can claim that $Q^l$ is determined by $P_1, \ldots, P_l$. Conditioned on $P_1, \ldots, P_l$, the partition of $[d] \setminus \bigcup_{i \in [l]} P_i$ is uniformly random. We consider $\{0, 1\}$-random variable $Y_j$, $j \in [d] \setminus \bigcup_{i \in [l]} P_i$. We represent $X^i(\mathbf{x})$ as a linear function of $Y_i$s as $X^i(\mathbf{x}) = \sum_{j \in [d] \setminus \bigcup_{i \in [l]} P_i} Y_j \mathbf{x}_j$ such that $Y_i = 1$ if $Y_i \in P_i$ and $Y_i = 0$ otherwise. By the concentration of linear functions over the Boolean slice (Theorem A.3), and recall $t = 8M\sqrt{\alpha \log d}$, we have

$$\mathbb{P}_{\mathcal{P}}\left[|X^i(\mathbf{x}) - \mathbb{E}[X^i(\mathbf{x})]| \geq \frac{t}{2}\right] \leq 2\exp\left(-\frac{t^2}{16(2M)^2}\right)$$
$$= 2\exp\left(-\frac{64M^2 \alpha \log d}{64M^2}\right)$$
$$= 2\exp\left(-\alpha \log d\right) = 2d^{-\omega(1)}.$$

Similarly, $\mathbb{P}\left[|X^{i+1}(\mathbf{x}) - \mathbb{E}[X^{i+1}(\mathbf{x})]| \geq \frac{t}{2}\right] \leq 2d^{-\omega(1)}$. Combining the fact that $\mathbb{E}[X^i(\mathbf{x})] = \mathbb{E}[X^{i+1}(\mathbf{x})]$, we have with probability at least $1 - d^{-\omega(1)}$, for any fixed $i \geq l$

$$\max\left\{|X^i(\mathbf{x}) - X^{i+1}(\mathbf{x})| - t, 0\right\} = 0,$$

which implies $g_{\mathcal{P}}(\mathbf{x}) = g_{\mathcal{P}}^l(\mathbf{x})$ with a probability at least $1 - rd^{-\omega(1)}$.

**Case 2:** $\|\mathbf{x}\| \geq 2M$. We have

$$g_{\mathcal{P}}(\mathbf{x}) = \|\mathbf{x}\| - M = g_{\mathcal{P}}^l(\mathbf{x}).$$

Combining these two cases, we have for any fixed query $\mathbf{x}$, with a probability at least $1 - rd^{-\omega(1)}$, we have $g_{\mathcal{P}}(\mathbf{x}) = g_{\mathcal{P}}^l(\mathbf{x})$.

By union bound over all queries $\mathbf{x} \in Q^l$, conditioned on that $\mathcal{E}_1, \ldots, \mathcal{E}_{l-1}$ occur, with probability at least $1 - r\mathsf{poly}(d)d^{-\omega(1)}$, $\mathcal{E}_l$ occurs. Therefore by induction,

$$P(\mathcal{E}_l) = P(\mathcal{E}_l|\mathcal{E}_1, \ldots, \mathcal{E}_{l-1})P(\mathcal{E}_{l-1}|\mathcal{E}_1, \ldots, \mathcal{E}_{l-2}) \ldots P(\mathcal{E}_2|\mathcal{E}_1)P(\mathcal{E}_1)$$
$$\geq 1 - r^2\mathsf{poly}(d)d^{-\omega(1)} = 1 - d^{-\omega(1)}.$$

This implies that with high probability, the computation path in round $l$ is determined by $P_1, \ldots, P_{l-1}$. Consequently, for all $l \in [\tau]$ a solution returned after $l - 1$ rounds is determined by $P_1, \ldots, P_{l-1}$ with high probability. By the same concentration argument, the solution is with a probability at least $1 - d^{-\omega(1)}$ in the form

$$(x_1, \ldots, x_\tau, x_\tau, \ldots, x_\tau),$$

up to an additive error $\mathcal{O}(t/2)$ in each coordinate.

Finally, we note that by allowing the algorithm to use random bits, the results are a convex combination of the bounds above, so the same high-probability bounds are satisfied. $\square$

Now we are ready to prove Theorem 4.5.

**Verification of $f_\mathcal{P}$.** Since $g_\mathcal{P}$ is 1-Lipschitz and convex, $g_\mathcal{P} + \frac{1}{2}\left\|\cdot\right\|^2$ is 1-Lipschitz and 1-strongly convex.

**Bound of total variation distance.** We first estimate the normalizing constant as follows.

$$
\begin{aligned}
Z_{f_\mathcal{P}} &= \int_{\mathbb{R}^d} \exp\left(-g_\mathcal{P}(\mathbf{x}) - \frac{1}{2}\left\|\mathbf{x}\right\|^2\right)\mathrm{d}\mathbf{x} \\
&\le \int_{\mathbb{R}^d} \exp\left(-\left\|\mathbf{x}\right\| + M - \frac{1}{2}\left\|\mathbf{x}\right\|^2\right)\mathrm{d}\mathbf{x} && \text{(by definition of } g_\mathcal{P}) \\
&\le \int_{\mathbb{R}^d} \exp\left(M - \frac{1}{2}\left\|\mathbf{x}\right\|^2\right)\mathrm{d}\mathbf{x} \\
&= \exp(M)\frac{2\Gamma^d(1/2)}{\Gamma(d/2)}\int_0^\infty t^{d-1}\exp(-t^2/2)\mathrm{d}t = \exp(M)\cdot(2\pi)^{d/2}.
\end{aligned}
\tag{6}
$$

Consider a subset $S = \left\{\mathbf{x}\in\mathbb{R}^d : \left\|\mathbf{x}\right\| \le M, X_{r+1} - X_r \ge t\right\}$. By Lemma C.1, with probability $1 - d^{-\omega(1)}$ over $\mathcal{P}$,

$$\rho(\mathsf{A}[f_\mathcal{P}, \mathbf{x}^0, r])(S) = 0.$$

Also, by definition of $f_\mathcal{P}$ and $g_\mathcal{P}$, we have for any $\mathbf{x}\in S$,

$$f_\mathcal{P}(\mathbf{x}) = g_\mathcal{P}(\mathbf{x}) + \frac{1}{2}\left\|\mathbf{x}\right\|^2 \le \frac{1}{2}\left\|\mathbf{x}\right\| + \frac{1}{2}\left\|\mathbf{x}\right\|^2 \le \frac{M + M^2}{2} \le M^2.$$

Thus, we have

$$\pi_{f_\mathcal{P}}(S) = \frac{\int_S \exp(f_\mathcal{P}(\mathbf{x}))\mathrm{d}\mathbf{x}}{Z_{f_\mathcal{P}}} \ge \frac{\int_S \exp(-M^2)\mathrm{d}\mathbf{x}}{Z_{f_\mathcal{P}}} = \frac{|S|\exp(-M^2)}{Z_{f_\mathcal{P}}}.$$

Recall $t = 8(M+1)\sqrt{\alpha\log d}$, we define the height of the sphere cap as

$$h = M - \frac{t}{\sqrt{2d_0}} = M - 8(M+1)\sqrt{\frac{\alpha\log d}{2d_0}}.$$

Let $b = (2Mh - h^2)/M^2 = 1 - \left(1 + \frac{1}{M}\right)^2\frac{64\alpha\log d}{d_0}$. By Lemma A.4 and Eq. equation 6, we have

$$
\begin{aligned}
\pi_{f_\mathcal{P}}(S) &\ge \frac{|S|\exp(-M^2)}{Z_{f_\mathcal{P}}} \\
&\ge \frac{|S|}{V_d M^d}\cdot V_d \cdot \frac{1}{\exp(M)\cdot(2\pi)^{d/2}}\cdot M^d\exp(-M^2) \\
&\ge \frac{|S|}{V_d M^d}\cdot V_d \cdot \frac{1}{(2\pi)^{d/2}}\cdot M^d\exp(-2M^2) \\
&= \frac{I_b\left(\frac{d+1}{2}, \frac{1}{2}\right)}{2}\cdot\frac{1}{\Gamma(d/2+1)}\cdot\frac{1}{2^{d/2}}\cdot\left(\frac{d}{4}\right)^{d/2}\exp\left(-d/2\right).
\end{aligned}
$$

by taking $M^2 = d/4$. Similarly, we have

$$\pi_{f_\mathcal{P}}(S) \ge \frac{\sqrt{\frac{t}{1-t}}}{2\pi\sqrt{d}(d+2)}\left(\frac{\sqrt{t}}{2}\right)^d.$$

Thus, there exists a constant $c \in (0, \frac{1}{2e})$ such that with probability $1 - d^{-\omega(1)}$ over $\mathcal{P}$,

$$\mathsf{TV}(\rho(\mathsf{A}[f_\mathcal{P}, \mathbf{x}^0, r]), \pi_{f_\mathcal{P}}) \ge \Omega(c^d).$$

# D PROOF OF THEOREM 5.1

## D.1 PROOF OF SMOOTH CASE

We define the hardness functions $f_\mathcal{P} : [-1, 1]^d \to \mathbb{R}$ as, $f_\mathcal{P}(\mathbf{x}) = S_1[g_\mathcal{P}](\mathbf{x})$ where $g_\mathcal{P} : \mathbb{R}^d \to \mathbb{R}$ is defined as

$$g_\mathcal{P}(\mathbf{x}) = L \cdot \left( |X^1| + \sum_{i \in [r]} \max\left\{ |X^i - X^{i+1}| - t, 0 \right\} \right)$$

with $t = 2\sqrt{d_0 + 2\sqrt{2} + 1}\sqrt{\alpha \log d}$ with $\alpha = \omega(1)$, $\alpha = \mathcal{O}(d^{1/3})$ and $L = \frac{1}{2\sqrt{d}}$.

Similarly, we have the following characterization of the output.

**Lemma D.1.** *For any randomized algorithm* A, *any* $\tau \leq r$, *and any initial point* $\mathbf{x}^0$, $X(\mathsf{A}[f_\mathcal{P}, \mathbf{x}^0, \tau])$ *takes form as*

$$(x_1, \ldots, x_\tau, x_\tau, \ldots, x_\tau),$$

*up to addictive error* $\mathcal{O}(t/2)$ *with probability* $1 - d^{-\omega(1)}$ *over* $\mathcal{P}$.

*Proof of Lemma D.1.* We fixed $\tau$ and prove the following by induction for $l \in [\tau]$: With high probability, the computation path of the (deterministic) algorithm A and the queries it issues in the $l$-th round are determined by $P_1, \ldots, P_{l-1}$.

As a first step, we assume the algorithm is deterministic by fixing its random bits and choose the partition of $\mathcal{P}$ uniformly at random.

To prove the inductive claim, let $\mathcal{E}_l$ denote the event that for any query $\mathbf{x}$ issued by A in iteration $l$, the answer is in the form $S_1[g_\mathcal{P}^l](\mathbf{x})$, where $g_\mathcal{P}^l : \mathbb{R}^d \to \mathbb{R}$ is defined as

$$g_\mathcal{P}^l = L \cdot \left( |X^1| + \sum_{i \in [l-1]} \max\left\{ |X^i - X^{i+1}| - t, 0 \right\} \right),$$

i.e., $\mathcal{E}_l$ represents the events that $\forall \mathbf{x} \in Q^l$, $f_\mathcal{P}(\mathbf{x}) = S_1[g_\mathcal{P}](\mathbf{x}) = S_1[g_\mathcal{P}^l](\mathbf{x})$.

Since the queries in round $l$ depend only on $P_1, \ldots, P_{l-1}$, if $\mathcal{E}_l$ occurs, the entire computation path in round $l$ is determined by $P_1, \ldots, P_l$. By induction, we conclude that if all of $\mathcal{E}_1, \ldots, \mathcal{E}_l$ occur, the computation path in round $l$ is determined by $P_1, \ldots, P_l$.

Now we analysis the conditional probability $P[\mathcal{E}_l \mid \mathcal{E}_1, \ldots, \mathcal{E}_{l-1}]$. By the property 3 of Theorem 4.2, $S_1[g_\mathcal{P}](\mathbf{x})$ only depends on the $\{g_\mathcal{P}(\mathbf{x}) : \mathbf{x}' \in B_1(\mathbf{x})\}$. Thus, it is sufficient to analyze the probability of the event that for a fixed query $\mathbf{x}$, any point $\mathbf{x}' \in B_1(\mathbf{x})$ satisfies that $g_\mathcal{P}(\mathbf{x}') = g_\mathcal{P}^l(\mathbf{x}')$. Given all of $\mathcal{E}_1, \ldots, \mathcal{E}_{l-1}$ occur so far, we can claim that $Q^l$ is determined by $P_1, \ldots, P_l$. Conditioned on $P_1, \ldots, P_l$, the partition of $[d] \setminus \bigcup_{i \in [l]} P_i$ is uniformly random. We consider $\{0, 1\}$-random variable $Y_j$, $j \in [d] \setminus \bigcup_{i \in [l]} P_i$. We represent $X^i(\mathbf{x}')$ as a linear function of $Y_i$s as $X^i(\mathbf{x}) = \sum_{j \in [d] \setminus \bigcup_{i \in [l]} P_i} Y_j \mathbf{x}'_j$

such that $Y_i = 1$ if $Y_i \in P_i$ and $Y_i = 0$ otherwise. By the concentration of linear functions over the Boolean slice (Theorem A.3), and recall $t = 2\sqrt{d_0 + 2\sqrt{2} + 1}\sqrt{\alpha \log d}$, we have

$$\mathbb{P}_\mathcal{P}\left[ |X^i(\mathbf{x}') - \mathbb{E}[X^i(\mathbf{x}')]| \geq \frac{t}{2} \right] \leq 2\exp\left( -\frac{t^2}{32 \sum_{i=1}^{d_0} (\mathbf{x}_i^\downarrow)^2} \right)$$

$$\leq 2\exp\left( -\frac{4\left((d_0 + 2\sqrt{2}\delta + \delta^2)\alpha \log d\right)}{32(d_0 + 2\sqrt{2}\delta + \delta^2)} \right)$$

$$= 2\exp\left( -\frac{\alpha \log d}{8} \right) = 2d^{-\omega(1)}.$$

Similarly, $\mathbb{P}\left[\left|X^{i+1}(\mathbf{x}') - \mathbb{E}[X^{i+1}(\mathbf{x}')]\right| \geq \frac{t}{2}\right] \leq 2d^{-\omega(1)}$. Combining the fact that $\mathbb{E}[X^i(\mathbf{x}')] = \mathbb{E}[X^{i+1}(\mathbf{x}')]$, we have with probability at least $1 - d^{-\omega(1)}$, for any fixed $i \geq l$

$$\max\left\{\left|X^i(\mathbf{x}') - X^{i+1}(\mathbf{x}')\right| - t, 0\right\} = 0,$$

which implies with a probability at least $1 - rd^{-\omega(1)}$, any point $\mathbf{x}' \in B_1(\mathbf{x})$ satisfies that $g_{\mathcal{P}}(\mathbf{x}') = g_{\mathcal{P}}^l(\mathbf{x}')$.

By union bound over all queries $\mathbf{x} \in Q^l$, conditioned on that $\mathcal{E}_1, \ldots, \mathcal{E}_{l-1}$ occur, with probability at least $1 - r\mathsf{poly}(d)d^{-\omega(1)}$, $\mathcal{E}_l$ occurs. Therefore by induction,

$$
\begin{aligned}
P(\mathcal{E}_l) &= P(\mathcal{E}_l|\mathcal{E}_1, \ldots, \mathcal{E}_{l-1})P(\mathcal{E}_{l-1}|\mathcal{E}_1, \ldots, \mathcal{E}_{l-2})\ldots P(\mathcal{E}_2|\mathcal{E}_1)P(\mathcal{E}_1) \\
&\geq 1 - r^2\mathsf{poly}(d)d^{-\omega(1)} = 1 - d^{-\omega(1)}.
\end{aligned}
$$

This implies that with high probability, the computation path in round $l$ is determined by $P_1, \ldots, P_{l-1}$. Consequently, for all $l \in [\tau]$ a solution returned after $l - 1$ rounds is determined by $P_1, \ldots, P_{l-1}$ with high probability. By the same concentration argument, the solution is with a probability at least $1 - d^{-\omega(1)}$ in the form

$$(x_1, \ldots, x_\tau, x_\tau, \ldots, x_\tau),$$

up to an additive error $\mathcal{O}(t/2)$ in each coordinate.

Finally, we note that by allowing the algorithm to use random bits, the results are a convex combination of the bounds above, so the same high-probability bounds are satisfied. $\square$

Now we are ready to prove the smooth case of Theorem 5.1.

**Verification of $f_{\mathcal{P}}$.** $g_{\mathcal{P}}$ is convex and 1-Lipschitz. By Theorem 4.2, $f_{\mathcal{P}}$ is convex and 1-smooth.

**Bound of total variation distance.** We first estimate the normalizing constant as follows. By property 1 of Theorem 4.2, we have

$$Z_{f_{\mathcal{P}}} = \int_{[-1,1]^d} \exp(-S_1[g_{\mathcal{P}}])\mathrm{d}\mathbf{x} \leq \int_{[-1,1]^d} \exp(-g_{\mathcal{P}} + 1)\mathrm{d}\mathbf{x} \leq e\int_{[-1,1]^d} \mathrm{d}\mathbf{x} \leq e \cdot 2^d.$$

Consider a subset $S = \left\{\mathbf{x} \in [-1,1]^d : |X^i| \leq \frac{t}{2}, \forall i \in [r], \frac{3t}{2} \leq X^{r+1} \leq \frac{t}{2} + t\sqrt{\alpha}\right\}$. By Lemma D.1, with probability $1 - d^{-\omega(1)}$ over $\mathcal{P}$,

$$\rho(\mathsf{A}[f_{\mathcal{P}}, \mathbf{x}^0, r])(S) = 0.$$

On the other hand, for any $\mathcal{P}$, recall $t = 2\sqrt{d_0 + 2\sqrt{2} + 1}\sqrt{\alpha \log d}$, $\alpha = \mathcal{O}(d^{1/3})$ and $L = \frac{1}{2d^{1/2}}$, we have

$$f_{\mathcal{P}}(\mathbf{x}) \leq \left(\frac{3t}{2} + t\sqrt{\alpha}\right)L \leq 4\alpha\sqrt{d_0 \log d}L < 1.$$

for sufficient large $d$. Thus, we have

$$\pi_{f_{\mathcal{P}}}(S) = \frac{\int_S \exp(-f(\mathbf{x}))\mathrm{d}\mathbf{x}}{Z_{f_{\mathcal{P}}}} \geq \frac{\int_S \exp(-1)\mathrm{d}\mathbf{x}}{Z_{f_{\mathcal{P}}}} \geq \frac{|S|}{e^2 2^d}.$$

By Lemma A.2, similarly, we have

$$\frac{|S|}{2^d} \geq c_1\exp(c_2\alpha \log d).$$

Thus, with probability $1 - d^{-\omega(1)}$ over $\mathcal{P}$,

$$\mathsf{TV}(\rho(\mathsf{A}[f, \mathbf{x}^0, r]), \pi_{f_{\mathcal{P}}}) \geq \Omega(d^{-\omega(1)}).$$

## D.2 PROOF OF LIPSCHITZ CASE

We define the hardness functions $f_{\mathcal{P}} : [-1,1]^d \to \mathbb{R}$ as:

$$f_{\mathcal{P}}(\mathbf{x}) \;=\; L \cdot \left( \left|X^1\right| + \sum_{i \in [r]} \max\left\{ \left|X^i - X^{i+1}\right| - t, 0 \right\} \right)$$

with $t = 2\sqrt{\alpha d_0 \log d}$ with $\alpha = \omega(1)$, $\alpha = \mathcal{O}(d^{1/3})$ and $L = \frac{1}{2\sqrt{d}}$.

Similarly, we have the following characterization of the output.

**Lemma D.2.** *For any randomized algorithm* A*, any* $\tau \leq r$*, and any initial point* $\mathbf{x}^0$*,* $X(\mathsf{A}[f_{\mathcal{P}}, \mathbf{x}^0, \tau])$ *takes form as*

$$(x_1, \ldots, x_\tau, x_\tau, \ldots, x_\tau),$$

*up to addictive error* $\mathcal{O}(t/2)$ *with probability* $1 - d^{-\omega(1)}$ *over* $\mathcal{P}$*.*

*Proof of Lemma D.2.* We fixed $\tau$ and prove the following by induction for $l \in [\tau]$: With high probability, the computation path of the (deterministic) algorithm A and the queries it issues in the $l$-th round are determined by $P_1, \ldots, P_{l-1}$.

As a first step, we assume the algorithm is deterministic by fixing its random bits and choose the partition of $\mathcal{P}$ uniformly at random.

To prove the inductive claim, let $\mathcal{E}_l$ denote the event that for any query $\mathbf{x}$ issued by A in iteration $l$, the answer is in the form

$$L \cdot \left( \left|X^1\right| + \sum_{i \in [l-1]} \max\left\{ \left|X^i - X^{i+1}\right| - t, 0 \right\} \right),$$

i.e., $\mathcal{E}_l$ represents the events that $\forall \mathbf{x} \in Q^l$, $f_{\mathcal{P}}(\mathbf{x}) = L \cdot \left( \left|X^1\right| + \sum_{i \in [l-1]} \max\left\{ \left|X^i - X^{i+1}\right| - t, 0 \right\} \right)$.

Since the queries in round $l$ depend only on $P_1, \ldots, P_{l-1}$, if $\mathcal{E}_l$ occurs, the entire computation path in round $l$ is determined by $P_1, \ldots, P_l$. By induction, we conclude that if all of $\mathcal{E}_1, \ldots, \mathcal{E}_l$ occur, the computation path in round $l$ is determined by $P_1, \ldots, P_l$.

Now we analysis the conditional probability $P[\mathcal{E}_l \mid \mathcal{E}_1, \ldots, \mathcal{E}_{l-1}]$.

Given all of $\mathcal{E}_1, \ldots, \mathcal{E}_{l-1}$ occur so far, we can claim that $Q^l$ is determined by $P_1, \ldots, P_l$. Conditioned on $P_1, \ldots, P_l$, the partition of $[d] \setminus \bigcup_{i \in [l]} P_i$ is uniformly random. We consider $\{0, 1\}$-random variable $Y_j, j \in [d] \setminus \bigcup_{i \in [l]} P_i$. We represent $X^i(\mathbf{x})$ as a linear function of $Y_i$s as $X^i(\mathbf{x}) = \sum_{j \in [d] \setminus \bigcup_{i \in [l]} P_i} Y_j \mathbf{x}_j$

such that $Y_i = 1$ if $Y_i \in P_i$ and $Y_i = 0$ otherwise. By the concentration of linear functions over the Boolean slice (Theorem A.1), and recall $t = 2\sqrt{\alpha d_0 \log d}$, we have

$$\mathbb{P}_{\mathcal{P}}\left[ |X^i(\mathbf{x}) - \mathbb{E}[X^i(\mathbf{x})]| \geq \frac{t}{2} \right] \;\leq\; 2\exp\left( -\frac{t^2}{32 d_0} \right)$$

$$= 2\exp\left( -\frac{4\alpha d_0 \log d}{32 d_0} \right)$$

$$= 2\exp\left( -\frac{\alpha \log d}{8} \right) = 2d^{-\omega(1)}.$$

Similarly, $\mathbb{P}\left[ |X^{i+1}(\mathbf{x}) - \mathbb{E}[X^{i+1}(\mathbf{x})]| \geq \frac{t}{2} \right] \leq 2d^{-\omega(1)}$. Combining the fact that $\mathbb{E}[X^i(\mathbf{x})] = \mathbb{E}[X^{i+1}(\mathbf{x})]$, we have with probability at least $1 - d^{-\omega(1)}$, for any fixed $i \geq l$

$$\max\left\{ \left|X^i(\mathbf{x}) - X^{i+1}(\mathbf{x})\right| - t, 0 \right\} = 0,$$

which implies $f_{\mathcal{P}}(\mathbf{x}) = L \cdot \left(|X^1| + \sum_{i \in [l-1]} \max\{|X^i - X^{i+1}| - t, 0\}\right)$ with a probability at least $1 - rd^{-\omega(1)}$.

By union bound over all queries $\mathbf{x} \in Q^l$, conditioned on that $\mathcal{E}_1, \ldots, \mathcal{E}_{l-1}$ occur, with probability at least $1 - r\mathsf{poly}(d)d^{-\omega(1)}$, $\mathcal{E}_l$ occurs. Therefore by induction,

$$P(\mathcal{E}_l) = P(\mathcal{E}_l | \mathcal{E}_1, \ldots, \mathcal{E}_{l-1})P(\mathcal{E}_{l-1} | \mathcal{E}_1, \ldots, \mathcal{E}_{l-2}) \ldots P(\mathcal{E}_2 | \mathcal{E}_1)P(\mathcal{E}_1)$$
$$\geq 1 - r^2\mathsf{poly}(d)d^{-\omega(1)} = 1 - d^{-\omega(1)}.$$

This implies that with high probability, the computation path in round $l$ is determined by $P_1, \ldots, P_{l-1}$. Consequently, for all $l \in [\tau]$ a solution returned after $l - 1$ rounds is determined by $P_1, \ldots, P_{l-1}$ with high probability. By the same concentration argument, the solution is with a probability at least $1 - d^{-\omega(1)}$ in the form

$$(x_1, \ldots, x_\tau, x_\tau, \ldots, x_\tau),$$

up to an additive error $\mathcal{O}(t/2)$ in each coordinate.

Finally, we note that by allowing the algorithm to use random bits, the results are a convex combination of the bounds above, so the same high-probability bounds are satisfied. $\qquad\square$

Now we are ready to prove the Lipschitz case of Theorem 5.1.

**Verification of $f_{\mathcal{P}}$.** $f_{\mathcal{P}}$ is convex and 1-Lipschitz.

**Bound of total variation distance.** We first estimate the normalizing constant as follows.

$$Z_{f_{\mathcal{P}}} \leq \int_{[-1,1]^d} \mathrm{d}\mathbf{x} = 2^d.$$

Consider a subset $S = \left\{\mathbf{x} \in [-1,1]^d : |X^i| \leq \frac{t}{2}, \forall i \in [r], \frac{3t}{2} \leq X^{r+1} \leq \frac{t}{2} + t\sqrt{\alpha}\right\}$. By Lemma D.2, with probability $1 - d^{-\omega(1)}$ over $\mathcal{P}$,

$$\rho(\mathsf{A}[f_{\mathcal{P}}, \mathbf{x}^0, r])(S) = 0.$$

On the other hand, for any $\mathcal{P}$, recall $t = 2\sqrt{\alpha d_0 \log d}$, $\alpha = \mathcal{O}(d^{1/3})$ and $L = \frac{1}{2d^{1/2}}$, we have

$$f_{\mathcal{P}}(\mathbf{x}) \leq \left(\frac{3t}{2} + t\sqrt{\alpha}\right)L \leq 4\alpha\sqrt{d_0 \log d}L < 1.$$

for sufficient large $d$. Thus, we have

$$\pi_{f_{\mathcal{P}}}(S) = \frac{\int_S \exp(-f(\mathbf{x}))\mathrm{d}\mathbf{x}}{Z_{f_{\mathcal{P}}}} \geq \frac{\int_S \exp(-1)\mathrm{d}\mathbf{x}}{Z_{f_{\mathcal{P}}}} \geq \frac{|S|}{e2^d}.$$

By Lemma A.2, we have

$$\frac{|S|}{2^d} \geq \left(c_0 \cdot \exp\left(-978\frac{\frac{9t^2}{16}}{d_0}\right) - \exp\left(-2\frac{\frac{t^2\alpha}{4}}{d_0}\right)\right)\left(1 - 2\exp\left(-\frac{t^2}{16d_0}\right)\right)^r$$
$$= \left(c_0 \cdot \exp\left(-\frac{4401}{2}\alpha \log d\right) - \exp\left(-2\alpha^2 \log d\right)\right)\left(1 - 2\exp\left(-\frac{\alpha \log d}{4}\right)\right)^r$$
$$\geq c_1 \exp(c_2\alpha \log d).$$

The last inequity holds since $\exp(-2\alpha^2 \log d) = O\left(c_0 \cdot \exp\left(-\frac{4401}{2}\alpha \log d\right)\right)$ and $r \leq d$ while $2\exp\left(\frac{\alpha \log d}{4}\right) = \Omega(d^\alpha)$. Thus, with probability $1 - d^{-\omega(1)}$ over $\mathcal{P}$,

$$\mathsf{TV}(\rho(\mathsf{A}[f, \mathbf{x}^0, r]), \pi_{f_{\mathcal{P}}}) \geq \Omega(d^{-c_2\alpha}).$$

# E UPPER BOUNDS

## E.1 UPPER BOUND FOR LOG-CONCAVE SAMPLING

In this section, we apply the algorithms in Fan et al. (2023) to our setting, which is summarized in Theorem E.1.

**Theorem E.1** (**Upper bound for very high accurate and weakly log-concave samplers (Proposition 4 Fan et al. (2023)))**. *For any uniform constant $c \in (0, 1/(2e))$, if an initial point $\mathbf{x}^0 \sim \rho_0$ satisfies $\chi^2_\pi(\rho_0) = \mathcal{O}(\exp(d))$, we can find a random point $x_T$ that has $c^d$ total variation distance to $\pi$ in*

- *$T = \tilde{O}\left(\mathsf{m}_2 d^{5/2}\right)$ steps if $\pi$ is 1-log-smooth and weakly log-concave;*

- *$T = \tilde{O}\left(\mathsf{m}_2 d^2\right)$ steps if $\pi$ is 1-log-Lipschitz and weakly log-concave,*

*where, $\mathsf{m}_2$ denotes the second moment. Furthermore, each step accesses only $\mathcal{O}(1)$ many queries in expectation.*

To prove this Theorem, we first recall the definition of semi-smooth (Definition E.2), then state the results for weakly log-concave samplers (Theorem E.3). Finally, we applies recent estimates of the worst-case Poincaré constant for isotropic log-concave distributions (Theorem E.5).

**Definition E.2** (**semi-smooth**). *We say $f : \mathbb{R}^d \to \mathbb{R}$ is bounded from below and is $L_\alpha$-$\alpha$-semi-smooth, i.e., $f$ satisfies for all $u, v \in \mathbb{R}^d$,*

$$\|\partial f(u) - \partial f(v)\| \le L_\alpha \|u - v\|^\alpha,$$

*for $L_\alpha > 0$ and $\alpha \in [0, 1]$. Here $\partial f$ represents a subgradient of $f$. When $\alpha > 0$, this subgradient can be replaced by the gradient. This condition implies $f$ is $L_1$-smooth when $\alpha = 1$ and a Lipschitz function satisfies this with $\alpha = 0$.*

**Theorem E.3** (**Proposition 4 Fan et al. (2023)**). *Suppose $\pi \propto \exp(-f)$ satisfies $C_{\mathsf{PI}}$-$\mathsf{PI}$ and $f$ is 1-$\alpha$-semi-smooth. Let $\varepsilon \in (0, 1)$. Then we can find a random point $x_T$ that has $\varepsilon$ total variation distance to $\pi$ in*

$$T = O\left(\frac{d^{\frac{\alpha}{\alpha+1}}}{C_{\mathsf{PI}}} \cdot \log\left(\frac{d^{\frac{\alpha}{\alpha+1}}}{C_{\mathsf{PI}}\varepsilon}\right) \log\left(\frac{\chi^2_\pi(\mu_0)}{\varepsilon^2}\right)\right),$$

*steps. Furthermore, each step accesses only $\mathcal{O}(1)$ many $f(x)$ queries in expectation.*

**Definition E.4** (**Poincaré inequality**). *A probability distribution $\pi$ satisfies the Poincaré inequality (PI) with constant $C_{\mathsf{PI}} > 0$ if for any smooth bounded function $u : \mathbb{R}^d \to \mathbb{R}$, it holds that*

$$Var_\pi(u) \le \frac{1}{C_{\mathsf{PI}}}\mathbb{E}\left[\|\nabla u\|^2\right].$$

**Theorem E.5** (**Poincaré Inequality for log-concave distribution Klartag (2023)**). *If $d \ge 2$, then log-concave distribution $\pi$ satisfies $(C \log d \|\mathsf{Cov}(\pi)\|_{\mathsf{op}})^{-1}$-$\mathsf{PI}$, where $C > 0$ is a universal constant, where $\mathsf{Cov}(\pi)$ is the covariance matrix of distribution $\pi$ and $\|\mathsf{Cov}(\pi)\|_{\mathsf{op}}$ is its operator norm.*

**Remark E.6.** *$\|\mathsf{Cov}(\pi)\|_{\mathsf{op}}$ can be bounded by second moment as*

$$\|\mathsf{Cov}(\pi)\|_{\mathsf{op}} \le \mathsf{Tr}(\mathsf{Cov}(\pi)) = \mathbb{E}\left[\|X - \mathbb{E}X\|^2\right] \le \mathsf{m}_2.$$

## E.2 COMPOSITE SAMPLERS

In this section, we apply the algorithm in Fan et al. (2023) to composite samplers

**Theorem E.7** (**Implication of Proposition 6 Fan et al. (2023)**). *If $f = f_1 + f_2$ where $f_1$ is 1-strongly convex and 1-smooth, and $f_2$ is 1-Lipschitz, then for $\pi \propto \exp(-f)$, we can find a random point $x_T$ that has $\varepsilon$ total variation distance to $\pi$ in*

$$T = O\left(d^{1/2} \log\left(\frac{d^{1/2}}{\varepsilon}\right) \log\left(\frac{\sqrt{H_\pi(\mu_0)}}{\varepsilon}\right)\right)$$

*steps.*

