# OpenReview forum: "The adaptive complexity of parallelized log-concave sampling"
_ICLR.cc/2025/Conference — ICLR 2025 Conference Withdrawn Submission_

### Official Review · Reviewer_nyUR · 2024-11-02

**Soundness:** 3
**Presentation:** 2
**Contribution:** 2
**Rating:** 6
**Confidence:** 3

**Summary:**

This paper studies the round complexity of sampling from log-concave distributions $\propto e^{-f}$ in $\mathbb{R}^d$ given (zeroeth order) query access to $f$. For unconstrained sampling from strongly log-concave and log-smooth densities, the present work shows that achieving target accuracy which is exponentially small in $d$ requires $\tilde{\Omega}(d)$ many rounds. They also extend the lower bound in this paper to give an $\tilde{\Omega}(d)$ lower bound for box-constrained sampling of log-concave and log-smooth/Lipschitz densities to inverse super-polynomial error. These lower bounds are within a polynomial of the right answer, e.g. in the unconstrained setting Anari et al. previously gave a parallel algorithm with round complexity $O(\log^2(d/\epsilon))$, which is $O(d^2)$ for exponentially small $\epsilon$. While the query complexity of log-concave sampling has been studied previously for low-dimensional distributions and for sampling Gaussians to constant accuracy, this paper appears to be the first to study the orthogonal question of getting lower bounds in the high-accuracy regime.

At a high level, the lower bound is based on constructing a certain potential $f$ which is a smoothing of a certain function which, for bounded-norm inputs, behaves like the following piecewise linear function. It is parametrized by an unknown partition of the coordinates into blocks of polylogarithmic size, and given an input $\mathbf{x}$, the function adds to the weight of $\mathbf{x}$ on the first set in the partition the following sum. It computes the imbalance between the weight of $\mathbf{x}$ for each consecutive pair of consecutive sets in the partition and, if the imbalance exceeds some threshold $t$ for that pair, it adds to the running sum how much that imbalance exceeds $t$. So for inputs on which the weights of $\mathbf{x}$ over the sets in the partition are roughly equal, this potential outputs something close to the weight of $\mathbf{x}$ on just the first set of the partition.

This potential is designed in such a way that in the first $\tau$ rounds of any parallel algorithm, if one thinks of the subsets in the partition as randomly chosen in sequence, then with high probability over the randomness of the subsequent subsets of the partition, the queries up to that point reveal no information about what the subsequent subsets are. This effectively forces the algorithm to run for $\tilde{\Omega}(d)$ rounds before it can query parts of space on which the potential is non-negligibly different from the weight of $\mathbf{x}$ on the first subset of the partition.

**Strengths:**

Deriving good lower bounds on query complexity is a central question in the area of log-concave sampling, and this paper makes solid progress towards understanding the optimal parallel query complexity in a natural setting, namely the high-accuracy regime. The lower bound construction is a nice blend between the techniques used to show adaptive complexity bounds in submodular optimization and the information-theoretic approach used in prior work of Chewi et al on query lower bounds for sampling. Additionally, it is interesting that they can derive a linear-in-$d$ lower bound even in the relatively low-accuracy regime in the box-constrained setting.

**Weaknesses:**

While these results are only relevant in the setting where the dimension $d$ is quite large, it is not clear that the lower bounds really get at the fundamental question of why there should be a dimension dependence in the query complexity of log-concave sampling. In particular, the fact that the authors get a near-linear lower bound for the unconstrained sampling setting feels more like a by-product of the fact that they are working in the high-accuracy regime than of the fact that the distribution lives in high dimensions (in particular, their lower bound could be consistent with a world in which the true round complexity is $\tilde{\Theta}(\log(1/\epsilon))$). Admittedly the near-linear lower bound in the box-constrained setting applies in the regime of only $1/d^{\omega(1)}$ error, but the argument seems almost verbatim the same as the argument in the unconstrained setting, suggesting that the fact they get such a lower bound in the relatively low error regime is more a quirk of the constrained setting than something fundamental about dimension dependence for log-concave sampling.

**Questions:**

The writing could be clarified in various places, e.g:
- In Lemma 4.3, the terminology "up to addictive error $O(t/2)$" it is only clarified later in the proof to mean that every *coordinate* is off by at most $O(t/2)$ (also why is there a big-O with a factor of 1/2?)
- The definition of $X_i(\mathbf{x}')$ seems off, as the subscript $i$ is then used to index over $i\in[\ell]$ in $\cup_{i\in[\ell]} P_i$
- The terminology "concentration of conditioned Bernoullis" is a little confusing as it is unclear what is being conditioned on. A more standard phrasing could be "concentration of linear functions over the Boolean slice
- When "partition $\mathcal{P} = (P_1,\ldots,P_{r+1})$ over the ground set" was introduced, it was not clear to me that the ground set was referring to the set of coordinates of the Euclidean space in which the target distribution lives

---

> ### Author Response · Authors · 2024-11-17
>
> We thank the reviewer for their thoughtful and constructive feedback, as well as for recognizing our solid progress towards understanding the optimal parallel query complexity, and a nice blend between the techniques. Below, we respond to the key points raised in the review.
>
> **"...by-product of the fact that they are working in the high-accuracy regime than of the fact that the distribution lives in high dimensions..."**
>
> We totally agree that, due to the focus on high accuracy and high dimension, our construction cannot fully decouple the dependence on dimensionality and accuracy. For **query complexity**, in particular, direct application of our construction could lead to high-accuracy hardness even with O(1) queries.
>
> However, our focus is on **adaptive complexity**, where the challenge lies in hiding information while using a polynomial number of queries per round. For this, high accuracy appears necessary to establish hardness.  Regarding dimensionality, we leverage concentration phenomena in high dimensions to effectively hide information. Exploring lower-accuracy regimes for both query and adaptive complexity remains an interesting and open direction for future work.
>
> **"The writing could be clarified in various places..."**
>
> Thank you for pointing out areas for improvement in clarity. We have addressed the specific issues in the revised version, including:
>
> - Clarifying "up to additive error \eta" in Lemma 4.3 and remove $1/2$.
> - We will replace $X_i(x)$ with $X^i(x)$ throughout the manuscript
> - Rephrasing "concentration of conditioned Bernoullis" to standard terminology like "concentration of linear functions over the Boolean slice."
> - Clarifying that the"“ground set of partition" refers to the coordinates of the Euclidean space.

---

> > ### Comment · Reviewer_nyUR · 2024-12-03
> > **Thanks for the reply!**
> >
> > I remain positive about this work and will keep my score.

---

### Official Review · Reviewer_VDWB · 2024-11-04

**Soundness:** 3
**Presentation:** 2
**Contribution:** 3
**Rating:** 8
**Confidence:** 2

**Summary:**

This paper shows lower bounds for a number of different log-concave sampling problems.  Specifically, they show a lower bound of $\tilde{\Omega}(d)$ objective function evaluations for the problem of sampling from a unconstrained distribution.  They also show a lower bound of $\\tilde{\Omega}(d)$ objective function evaluations for the problem of sampling from a “box-constrained” logconcave distribution constrained to the unit cube.  In addition to applying to any unconstrained or box-constrained distribution, their lower bounds also handle the special cases when the log-density is smooth and Lipschitz.  However, they do not obtain lower bounds for the special case where the log-density is strongly convex.

**Strengths:**

The paper improves over previous works which show lower bounds for the problem of sampling from a logconcave distribution.

The comparison to previous works for upper bounds is very clearly stated.   However, the comparison to previous works on lower bounds is less clear, as there are different situations where such lower bounds apply (e.g., different accuracy levels) (see weaknesses below)

**Weaknesses:**

The writing in the paper could be improved in certain aspects.  In particular, the comparison to prior works is somewhat confusing and could be made more clear.

As mentioned above, the comparison to previous works for upper bounds is very clearly stated, with a clear table—which is good.      However, the comparison to previous works on lower bounds is less clear, as there are different situations where such lower bounds apply (e.g., different accuracy levels).  It would be good to include a side-by-side comparison with previous works on lower bounds, perhaps as an additional table in the appendix if there is not enough room in the main body of the paper.

Additionally, it would be helpful to have a more intuitive discussion of the difference between adaptive complexity vs. query complexity.  I am more familiar with query complexity, but adaptive complexity seems to be a less common term and it would be good to highlight the differences between the two concepts, and to explain with respect to which concepts the authors improve on previous lower bounds.

**Questions:**

Could the authors clarify in what settings/regimes they improve over lower bounds from prior works?   Under what assumptions on the objective function, and for what required accuracy levels were the improvements, and by how much was the improvement? In what regimes were lower bounds previously available/not available?  In what settings were they available, but improved by the authors?  In what regimes were they not improved?

Also, I understand that the authors results apply both to zeroth-order oracle and first-order oracles, which is good. However, do they improve on previous lower bounds for first-order oracles, or is the improvement only for zeroth-order oracles?

---

> ### Author Response · Authors · 2024-11-17
>
> Thank you for your comprehensive review and valuable feedback. Below, we address the specific points raised:
>
> **"...improve over previous works which show lower bounds... comparison to previous works on lower bounds... clarify in what settings/regimes they improve over lower bounds from prior works...do they improve on previous lower bounds for first-order oracles..."**
>
>
> We respectfully disagree with this point, as our work establishes **first lower bounds** for parallel log-concave sampling. Prior works primarily focused on upper bounds, and no previous results have addressed lower bounds in these settings.
>
>
> While we understand that the reviewer might have assumed the existence of prior lower bounds due to the parallel developments in related fields or query complexity of log-concave sampling, these are not the case for the specific problem of adaptive parallel sampling. Therefore, a direct comparison is technically not available. We will revise the manuscript to explicitly highlight that these are the first lower bounds for the settings we consider.
>
>
> **"... intuitive discussion of the difference between adaptive complexity vs. query complexity..."**
>
> Thank you for this suggestion. Adaptive complexity measures the number of interaction rounds needed in a parallel or distributed setting, whereas query complexity refers to the total number of oracle calls. This distinction is critical because reducing adaptive complexity can significantly improve runtime in parallel systems, even if the total query complexity remains unchanged. We will add a more intuitive explanation of this distinction in the manuscript.
>
> **"do they improve on previous lower bounds for first-order oracles, or is the improvement only for zeroth-order oracles?"**
>
> Since this work establishes the first lower bounds for parallel sampling in both zeroth-order and first-order oracle settings, the notion of “improvement” over prior lower bounds is not applicable here.

---

> ### Comment · Reviewer_VDWB · 2024-11-26
>
> Thank you for the clarification.  As your improvements are exclusively for parallelized samplers, I think that it would be good to change the title to "The adaptive complexity of parallelized log-concave sampling".  This should clear up the confusion for the reader.
>
> I have raised my score in light of the clarification, with the expectation that the title should be changed to highlight the parallel nature of the work.

---

> ### Author Response · Authors · 2024-11-27
>
> Thank you very much for your feedback. This encourages us a lot. Following your suggestion, we have updated the manuscript, with the title revised to "The adaptive complexity of parallelized log-concave sampling" in the updated PDF. We believe this should clear up the confusion for the reader. We also made minor updates to the language and presentation.
>
> We are more than happy to respond to any further questions and discussions.

---

### Official Review · Reviewer_MYcM · 2024-11-06

**Soundness:** 3
**Presentation:** 3
**Contribution:** 3
**Rating:** 8
**Confidence:** 3

**Summary:**

This paper studies the problem of parallel sampling, and shows a lower bound of $\widetilde O(d)$ for the number of parallel iterations necessary in the "high-accuracy" regime for log-concave sampling. The paper also shows a similar result for the box-constrained setting. To achieve these lower bounds, the paper leverages techniques from the optimization literature that have been used to show lower bounds for adaptive algorithms for optimization. The paper also studies some other sampling regimes such as composite sampling, showing similar bounds.

**Strengths:**

This paper studies an interesting problem, and shows how to leverage techniques from a different but related area to show lower bounds for parallel sampling. The lower bounds are of interest to both theoreticians and practitioners, since parallel sampling methods for diffusion models have recently been proposed and have gained popularity.

Generally, the paper is well-written and easy to understand. I recommend acceptance

**Weaknesses:**

It would be nice to see some results for diffusion models, since it seems like they should naturally follow from your results. It would also be nice to cite the recent works from the diffusion literature on parallel sampling (see [1, 2, 3]). This would make this work far more appealing to practitioners.

It would also be nice to have a thorough description of the barriers in extending your techniques to the low-accuracy regime, and any specific intermediate conjectures towards proving such results.

[1] https://arxiv.org/abs/2305.16317

[2] https://arxiv.org/abs/2406.00924

[3] https://arxiv.org/abs/2405.15986

**Questions:**

1) Can your results be extended to diffusion models in a direct way?
2) What can you say about the low-accuracy regime?

---

> ### Author Response · Authors · 2024-11-17
>
> We appreciate the reviewer’s thoughtful and constructive feedback, as well as the recommendation for acceptance. Below, we address the specific questions and suggestions raised:
>
> **"...nice to see some results for diffusion models...Can your results be extended to diffusion models in a direct way?.."**
>
> Thank you for this suggestion. While our current results focus on log-concave sampling, the connection to diffusion models is indeed intriguing and highly relevant for algorithm design. Specifically, many log-concave sampling methods are based on discretizing a dynamic that converges to the target distribution. Similarly, the inference process in diffusion models involves discretizing dynamics discribed by stochastic differential equations or probability flow ordinary differential equations for the reverse process.
>
> However, the sampling methods used in diffusion models are not solely based on dynamics, which differentiates the lower-bound analysis. To establish lower bounds for diffusion models, it may be useful to leverage tools about information-based complexity in the scientific computation community, which considers the complexity of discretizing differential equations.
>
> In the revised manuscript, we will add a discussion about the future potential of extending our lower-bound techniques to diffusion models. Additionally, we will cite recent works on parallel sampling for diffusion models ([1], [2], [3]) to better contextualize our contributions and highlight their relevance to practitioners.
>
> **"What can you say about the low-accuracy regime?"**
>
> In this study, we concentrate on scenarios involving high dimensions and very high accuracy. Extending our analysis to broader accuracy ranges presents significant challenges. Specifically, as outlined in Lemma 3.3 of our study, the set of points that remain unreachable by time $t$ is defined as ${x : |X_i - X_j| \geq t, \forall i, j \in [r+1] \text{ and } i,j \geq t+1}$. Accurately estimating the proportion of such an unreachable set presents significant difficulties. Furthemore, constructing an unreachable set with a large proportion of density using a polynomial number of queries is particularly challenging. We view this as an important direction for future research and will include this discussion in the revised manuscript to highlight the open questions related to the low-accuracy regime.

---

> > ### Comment · Reviewer_MYcM · 2024-11-27
> >
> > Thanks for answering my questions. I maintain my score.

---

### Official Review · Reviewer_niWg · 2024-11-11

**Soundness:** 2
**Presentation:** 2
**Contribution:** 2
**Rating:** 5
**Confidence:** 3

**Summary:**

This paper provides lower bounds for the problem of adaptive sampling from a distribution over $\mathbb{R}^d$ with probability density functions proportional to $exp(-f)$ for some smooth, Lipschitz or convex function $f:\mathbb{R}^d\rightarrow\mathbb{R}$. The algorithm can have access to a 0-th order oracle (i.e can query the values of $f$) that can also be translated to gradient queries with a polynomial blowup in the number of queries. Before making each query, the algorithm can also see the replies of the oracle to the previous queries. The lower bounds are also extended to box-constrained sampling. In the latter, the total variation distance between the true and the sampled distribution (sampling accuracy) is inverse polynomial in the dimension, while in the former case, it is inverse exponential. The lower bounds are linear on the dimension $d$ up to logarithmic factors, while the best known upper bounds are higher degree polynomial depending on the function $f$.

**Strengths:**

First attempt for a lower bound on adaptive sampling applicable to a wide range of distributions.

**Weaknesses:**

The paper only has negative results (lower bounds), which are not tight for any parameter regime.
The writeup could also be improved in terms of typos/grammar as well as clarity of the presentation (especially in section 3).



Minor comments:
Line 15: “minimal”->”minimum”
Line 21 and 100: “sup-”->”sub-”
Line 21: “small accuracy” sounds weird (because it means “high accuracy” here). Maybe “small error” would be better.
Line 85: Mention that c<1.
Section 3: There are multiple occasions where you are correctly describing a reduction FROM hypothesis testing TO sampling, which is indeed the way to translate a hypothesis testing lower bound to a samling lower bound. However, you incorrectly claim the reduction is the other way around, which is a bit confusing. (e.g see Lines 225-226, 235,269,274).

Line 267: “total variance”->”total variation”

**Questions:**

Can you comment on the possibility to close the gap between upper and lower bounds in some parameter regime?

---

> ### Author Response · Authors · 2024-11-17
>
> We are grateful that the reviewer acknowledges the novelty of our work as the **first lower bound** for adaptive sampling applicable to a wide range of distributions.
>
> **"...only has negative results (lower bounds), which are not tight..."**
>
> We respectfully disagree with the claim that our results are not tight for any parameter regime. In fact, our results are point-wise tight when compared to an anonymous work recently submitted to ICLR, which provides corresponding upper bounds for adaptive sampling [1].
>
> [1] https://openreview.net/forum?id=6Gb7VfTKY7&nesting=2&sort=date-desc
>
> **"The writeup could also be improved..."**
>
> We apologize for any lack of clarity or typographical issues in the current draft. We will carefully proofread the manuscript to address the specific issues raised by the reviewer.
>
> For Section 3, we will revise and restructure the explanations to ensure clarity, especially regarding the direction of the reductions.
>
> For the suggestion to replace "sup-" with "sub-" in Lines 21 and 100, we would like to point out that this is not a typo. The term “sup-polynomially small accuracy” is technically precise and refers to quantities smaller than any polynomial order of $d$.
>
> **"... the possibility to close the gap between upper and lower bounds..."**
>
> Thank you for this insightful question. Compared with the anonymous work [1], our lower bound highlights the pivotal role of $\log(1/\epsilon)$, which is essential for understanding adaptive sampling complexity. However, the dependence on d remains an open question and likely varies across distribution classes or oracle assumptions. Exploring sharper dimension dependent bounds is a key direction for future work, and we will elaborate on these open questions in the revised manuscript.

---

> ### Author Response · Authors · 2024-11-24
>
> Thank you again for reviewing the manuscript. In the author rebuttal, we have responded to the tightness issue by justifying the tightness of our lower bounds. This seems to be the main (and the only, apart from minor issues) concern raised by the reviewer. As the author-reviewer discussion period is approaching its end, we would like to know if you have any additional questions or concerns. We are happy to provide our response if so. At the moment, the rating of 3 seems very low without enough justification.

---

> > ### Comment · Reviewer_niWg · 2024-11-26
> >
> > Thank you for letting me know about the other paper submitted to ICLR. Given this information, I have increased my score to 5. I would not be against acceptance, if the clarity of the writeup is significantly improved and discussion on tightness (using this other ICLR submission) is added to it.

---

> > > ### Author Response · Authors · 2024-11-27
> > >
> > > Thank you again for your review and the follow-up discussion. Following your suggestion, we have updated the [manuscript](https://openreview.net/pdf?id=EeqlkPpaV8) to include the discussion on the matching upper/lower bounds. We have also spent time going through the manuscript to improve the presentation. There are around 30 minor changes made to polish the manuscript (highlighted in blue color). We would like to request you to re-evaluate the work given the updated manuscript, and we are more than happy to respond to any further questions and discussions.

---

> ### Author Response · Authors · 2024-11-29
>
> Thank you again for your review and the discussions. We believe our presentation has been improved by discussing the matching upper bound and by polishing the manuscript. As the author-reviewer discussion period is approaching its end, would you mind taking some time to look at the updated manuscript? We are happy to answer any additional questions.

---

### Note · Authors · 2025-12-01

**Comment:**

We withdraw this paper due to a critical error in the analysis that invalidates its main theoretical claims. In particular, Lemma 4.3 does
not imply the required total variation gap. Since this step is essential for the inductive bound used to establish the main results, the subsequent guarantees in the paper cannot be justified. We apologize for the oversight.

**Withdrawal Confirmation:**

I have read and agree with the venue's withdrawal policy on behalf of myself and my co-authors.

---

### Meta-Review · Area_Chair_ZGf9 · 2024-12-21

**Metareview:**

This paper considers the adaptive complexity of sampling, i.e. the minimum number of rounds to sample from a target with polynomially many queries in parallel at each round. Authors focus on total variation distance. In the unconstrained case, they study log-smooth, log-Lipschitz and log strongly or non-strongly concave distributions and provide negative results that prove a linear complexity algorithm cannot return a sample with  exponentially small error. In the constrained case, authors show that the same with sup-polynomially small error, for log-concave distributions.


This paper was reviewed by four expert reviewers the following Scores/Confidence: 6/3, 8/2, 8/3, 5/3. I think the paper is studying an interesting topic and the results are relevant to ICLR community. The following concerns were brought up by the reviewers:

- The paper has tight lower bounds. But I think authors should include a more detailed discussion on upper bounds in different settings as well. Authors should also clarify in their paper, point-wise tightness when compared to the work they cite in the rebuttal.

- several typos and ambiguous statements were pointed by the reviewers. These should be carefully addressed.


Authors should carefully go over reviewers' suggestions and address any remaining concerns in their final revision. Based on the reviewers' suggestion, as well as my own assessment of the paper, I recommend including this paper to the ICLR 2025 program.

**Additional Comments On Reviewer Discussion:**

Reviewer questions are thoroughly answered by the authors.

---

### Decision · Program_Chairs · 2025-01-22

Accept (Poster)